# Proving Olympiad Algebraic Inequalities without Human Demonstrations

**Chenrui Wei[1]**
chenruiw97@gmail.com

**Mengzhou Sun[2]**
sunm07@u.nus.edu

**Wei Wang[1, *]**
wangwei@bigai.ai

[1]State Key Laboratory of General Artificial Intelligence, BIGAI, Beijing, China
[2]Department of Mathematics, National University of Singapore

## Abstract

Solving Olympiad-level mathematical problems represents a significant advancement in machine intelligence and automated reasoning. Current machine learning methods, however, struggle to solve Olympiad-level problems beyond Euclidean plane geometry due to a lack of large-scale, high-quality datasets. The challenge is even greater in algebraic systems, which involve infinite reasoning spaces within finite conditions. To address these issues, we propose *AIPS*, an *Algebraic Inequality Proving System* capable of autonomously generating complex inequality theorems and effectively solving Olympiad-level inequality problems without requiring human demonstrations. During proof search in a mixed reasoning manner, a value curriculum learning strategy on generated datasets is implemented to improve proving performance, demonstrating strong mathematical intuitions. On a test set of 20 International Mathematical Olympiad-level inequality problems, AIPS successfully solved 10, outperforming state-of-the-art methods. Furthermore, AIPS automatically generated a vast array of non-trivial theorems without human intervention, some of which have been evaluated by professional contestants and deemed to reach the level of the International Mathematical Olympiad. Notably, one theorem was selected as a competition problem in a major city's 2024 Mathematical Olympiad. All the materials are available at *sites.google.com/view/aips2*.

## 1 Introduction

One of the key milestones in the field of artificial intelligence is the capability to reason (Pearl 1998) and prove theorems (Wu 1978; Chou et al. 2000; Trinh et al. 2024). However, theorem proving often involves long reasoning chains, complex mathematical structures, intricate calculations, and infinite reasoning spaces. Consequently, developing AI capable of proving complex mathematical theorems requires sophisticated reasoning and the ability to navigate through an extensive search space to construct a valid proof. The complexity of these problems lies in the need for effective heuristics and strategies to manage the vast number of possible actions and the lengthy sequences of logical steps necessary to arrive at a solution.

Existing work on grade school and college admission math problems has achieved notable success, e.g., GSM8K (Cobbe et al. 2021) and SAT Math (Achiam et al. 2023), which demonstrate better performance on tasks such as arithmetic and basic algebra. However, research focused on solving International Mathematical Olympiad (IMO)-level problems remains relatively sparse. Notable efforts in this area include AlphaGeometry (Trinh et al. 2024), and GPT-*f* (Polu and Sutskever 2020) on miniF2F (Zheng et al. 2021), which have made progress in solving Euclidean plane geometry at the Olympiad level and various mathematical competition problems, respectively.

---

*Corresponding author.

A significant challenge for learning-based methods in this domain is the scarcity of suitable datasets, which limits the ability to train models effectively and hampers progress in achieving human-level performance on these high-difficulty problems. The miniF2F dataset (Zheng et al. 2021) includes only 244 validation and 244 test mathematical problems from various competitions. AlphaGeometry (Trinh et al. 2024) addresses this issue by synthesizing millions of theorems and proofs across different levels of complexity to train a neural language model from scratch. Similarly, the INequality Theorem proving benchmark, INT (Wu et al. 2020), can synthesize a theoretically unlimited number of theorems and proofs in the domain of algebraic equalities and inequalities. However, INT focuses on testing a learning-assisted theorem proving agent's generalization ability rather than increasing the difficulty to competition level.

Another significant challenge in automated theorem proving is designing effective search strategies to navigate the vast space of possible proofs. Recent advancements have highlighted various approaches to enhance search efficiency and proof success rates. Some studies have shown that incorporating Monte Carlo Tree Search (MCTS) at test time can significantly aid in proving new theorems (Wu et al. 2020). Inspired by the success of AlphaZero (Zhang and Yu 2020), other research has explored HyperTree Proof Search (HTPS) (Lample et al. 2022), which learns from previous proof searches through online training, iteratively improving its strategy by learning which paths are more likely to lead to successful proofs. Another innovative approach starts the proof search from the root goal that needs to be proved (Polu and Sutskever 2020), expanding a maintained proof tree by prioritizing open goals based on their cumulative log probability.

In this work, we introduce *AIPS*, an *Algebraic Inequality Proving System*, which can generate a large number of high-quality theorems and solve IMO-level algebraic problems. AIPS focuses on ternary and quaternary inequalities, excluding $n$-variable inequalities represented recursively in formal verification systems. Among the generated theorems, some have proven to be very challenging, with one selected for a major city's 2024 Mathematical Olympiad. We present novel and challenging inequality theorems discovered by AIPS in the appendix, which have been carefully evaluated by IMO-level professional contestants and found to be comparable to IMO inequalities from around the year 2000.

Additionally, AIPS incorporates a value network to evaluate newly generated inequalities, selecting subgoal candidates based on the top scores provided by the value network. The value network is trained on synthetic datasets with increasing difficulty in a curriculum manner. In our experiments, AIPS proved difficult theorems up to the IMO level and solve 10 out of 20 problems in an IMO-level inequality test, significantly surpassing the performance of previous Large Language Model-based theorem provers (Polu and Sutskever 2020; Polu et al. 2022; Yang et al. 2024; Song et al. 2024).

The main contributions in this paper are summarized as follows:

1. We propose a symbolic deductive engine capable of efficiently generating high-quality and solving high-difficulty algebraic inequality theorems. This engine addresses the bottleneck of lacking large-scale, high-quality data in this field.

2. We demonstrate that a symbolic algebraic inequality prover can be significantly enhanced under the guidance of a value network, especially when the value network is trained in a curriculum manner.

3. Our AIPS can generate challenging and elegant inequality theorems, including one selected for a major city's Mathematical Olympiad. AIPS proves 10 out of 20 IMO-level inequalities, surpassing state-of-the-art methods and producing highly human-readable proofs.

## 2   Related Work

**Automated Theorem Proving.** Automated theorem proving has been a focus of artificial intelligence since the 1950s (Harrison et al. 2014; Wu 1978). Modern theorem provers, based on tactic and premise selection, search for proofs by interacting with proof assistants such as Lean (De Moura et al. 2015), Coq (Barras et al. 1999) and Isabelle (Nipkow et al. 2002). They struggle with the rapidly expanding search space and the scarcity of high-quality datasets in most mathematical domains. The challenge is even greater for proving algebraic inequalities, which involve complex computational rules. Previous efforts to address this issue have focused on augmenting tactic selection and premise prediction in interactive theorem provers (Polu and Sutskever 2020; Polu et al. 2022; Yang et al.

2024). However, these provers have only been able to solve problems of limited difficulty in this field. In this paper, our AIPS can solve highly complex algebraic inequality theorems up to the IMO level.

**Datasets and Benchmarks for Theorem Proving.** Formal mathematical libraries, such as Isarstep (Li et al. 2020), Mathlib (van Doorn et al. 2020), and CoqGym (Yang and Deng 2019), currently serve as the primary datasets for theorem proving. These libraries, manually curated by humans, include many intricate and profound proofs, such as the formal proofs of the Four-Color Theorem (Gonthier et al. 2008), the Liquid Tensor Experiment (Scholze 2022), and Fermat's Last Theorem (Buzzard and Taylor 2024). Due to the labor-intensive nature of manual proof writing, these libraries are relatively small, typically containing around 200,000 theorems. While they encompass a wide range of mathematical fields, the number of theorems in specific areas is quite limited.

Synthetic theorems can provide large-scale datasets for learning-based theorem provers (Polu and Sutskever 2020; Wu et al. 2020). However, these theorems are often of limited difficulty. Recently, significant progress has been made in synthesizing geometry theorems (Trinh et al. 2024) using neural theorem provers. In this paper, we develop AIPS for algebraic inequalities, which can automatically and efficiently generate a large number of intricate theorems, with some reaching the IMO level. These theorems will significantly improve neural theorem proving methods.

**Search Strategy for Efficient Inference.** Deep learning has achieved remarkable success in enhancing search algorithms (Silver et al. 2016, 2017). Proof search in theorem proving, however, is more challenging compared to self-play games like Go, as it may involve an infinite search space within finite conditions. INT (Wu et al. 2020) incorporates MCTS, while HyperTree Proof Search (HTPS) (Lample et al. 2022) employs online training to improve search strategy. GPT-*f* (Polu and Sutskever 2020) learns a value network to guide backward search. Our AIPS integrates the benefits of both HTPS and GPT-*f*, introducing a value curriculum learning strategy.

## 3 Algebraic Inequality Proving System

### 3.1 Symbolic Deductive Engine for Algebra

Interactive theorem provers, such as Lean, can verify mathematical operations but lack the ability to perform automated mathematical reasoning by combining computational rules. This challenge is amplified in the automatic proof of algebraic inequalities, which often involves numerous calculations, extensive transformation rules, and complex theorem matching. To address this, we design a symbolic deductive engine for algebra, encompassing dozens of fundamental theorems and transformation rules for algebraic inequalities. It integrates with the symbolic computation system SymPy [2], enabling effective algebraic reasoning.

#### 3.1.1 Representation for Algebraic Expressions and Theorems

Algebraic expressions are represented symbolically with an underlying expression tree structure as shown in Fig. 1. The basic computational rules include self-equivalence transformations of inequalities and various built-in SymPy functions, such as combining fractions (`sympy.together`) and expanding expressions (`sympy.expand`). Our deductive engine's library also includes fundamental algebraic inequality theorems: the Arithmetic Mean-Geometric Mean Inequality (AM-GM), the weighted AM-GM Inequality, Cauchy's Inequality, Jensen's Inequality, the discrete Hölder's Inequality, Schur's Inequality, the binary and ternary Muirhead's Theorem. Each inequality is represented as a category of theorem matching, containing variables, conditions, conclusions, and equality conditions.

#### 3.1.2 Pattern Matching for Inequality Theorems

During symbolic reasoning, the system attempts to apply inequality theorems to a particular algebraic expression or inequality, as shown in Fig. 1. When matching algebraic expressions with inequality theorems, it first traverses the expression tree to determine how the value of the entire expression changes as the node's value increases, updating the node's label accordingly. If the change cannot be determined, no theorem matching is performed on the subtree of that node. After completing the labeling, the system matches the next layer of determinable nodes with theorems. If a match

---

[2]https://www.sympy.org/

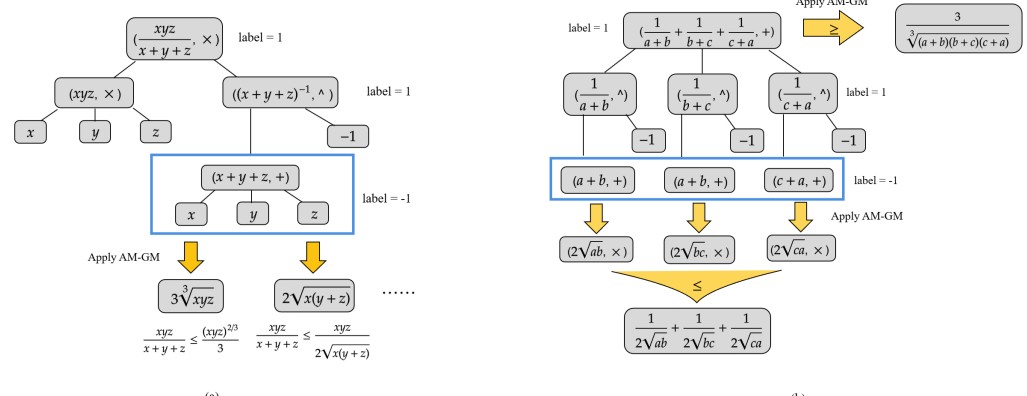

Figure 1: Examples of expression trees and pattern matching for the AM-GM inequality are illustrated. In (a), for $x, y, z \geq 0$, the value of $\frac{xyz}{x+y+z}$ decreases as $x + y + z$ increases, so the label of the node $x + y + z$ is $-1$. By applying the AM-GM inequality, we derive a series of upper bounds with respect to the root, e.g., $\frac{(xyz)^{2/3}}{3}$ and $\frac{xyz}{2\sqrt{x(y+z)}}$. In (b), when traversing the expression tree of $\frac{1}{a+b} + \frac{1}{b+c} + \frac{1}{c+a}$, pattern matching for the AM-GM inequality at various nodes yields different types of bounds, such as the upper bound $\frac{1}{2\sqrt{ab}} + \frac{1}{2\sqrt{bc}} + \frac{1}{2\sqrt{ca}}$ and the lower bound $\frac{3}{((a+b)(b+c)(c+a))^{\frac{1}{3}}}$.

is successful, the matched sub-expression is replaced with the new expression obtained using the theorem. Based on the previous labels, it then determines whether the entire expression increases or decreases, thereby deriving a new inequality. For certain inequality theorems, such as Jensen's Inequality, pattern matching is particularly complex and time-consuming. Therefore, to improve the efficiency of reasoning at each step, we have imposed time limits on the matching process for some theorems.

### 3.1.3 Forward Reasoning

Forward reasoning in theorem proving involves matching variables and conditions to a theorem and deducing new conclusions. In our engine, new inequalities can be obtained by matching theorems to both sides of an inequality or by applying self-equivalence transformation rules. If any two of the resulting inequalities can be connected (e.g., applying $a \leq b$ and $b \leq c$ to derive $a \leq c$), the system continues to link them to form new inequalities. Therefore, our engine has the capability to perform forward reasoning to generate large-scale data.

### 3.2 Olympiad-Level Inequality Proof Set

One of the main challenges in enabling learning-based models to solve complex mathematical problems is the scarcity of large-scale, high-quality datasets. To overcome this obstacle, we develop a theorem generator that effectively generates Olympiad-level inequality theorems by enhancing the methods described in Section 3.1.3.

### 3.2.1 Synthetic Theorem Generation

We randomly generate thousands of cyclically symmetric symbolic expressions, which serve as the initial premises for our reasoning process. Utilizing 32 CPUs, we run Algorithm 1 for 8 hours, resulting in the generation of 191,643 inequality theorems. The generated inequalities are stored in a tree structure, with each node containing the necessary information for extracting proofs and training machine learning models. Fig. 2 shows the procedure of generating a synthetic theorem in our AIPS, and Fig. 3(a) shows the distribution of inference depths in the generated inequalities.

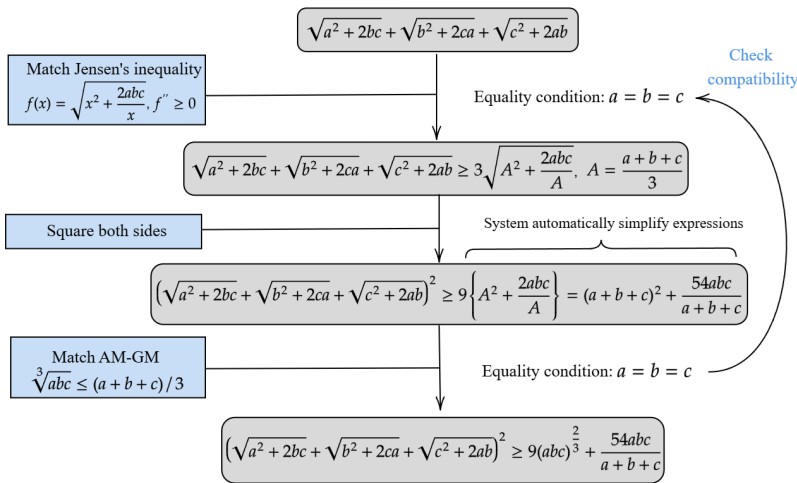

Figure 2: An example of generating synthetic theorems in AIPS. When the initial premise $\sqrt{a^2 + 2bc} + \sqrt{b^2 + 2ca} + \sqrt{c^2 + 2ab}$ successfully matches with Jensen's inequality, a new inequality is generated. By subsequently applying transformation rules and matching other fundamental inequalities, such as the AM-GM inequality, the deductive engine incrementally generates new inequality theorems. When an inequality theorem is applied, the system verifies whether the equality condition holds, e.g., $a = b = c$.

### 3.2.2 Synthetic Theorem Evaluation

To evaluate the quality of our dataset, we select 10 problems with reasoning lengths exceeding five steps, and invite two National Mathematical Olympiad gold medalists and one silver medalist to assess the difficulty and elegance of these problems. Their evaluations reveal that our dataset contains a vast array of non-trivial theorems, some of which surpass the difficulty of inequalities found in early IMO competitions. Notably, one inequality theorem from our dataset is selected for a major city's Mathematical Olympiad. All the 10 problems and evaluation results are provided in Appendix **??**.

## 3.3 Neural Algebraic Inequality Prover

By leveraging the capabilities of the deductive engine introduced in Section 3.1 and the Best-First-search algorithm (Dechter and Pearl 1985), we develop an algebraic inequality prover. This prover formulates the algebraic inequality proving as a sequential decision-making process by selecting theorems to generate highly human-readable proofs. As shown in Fig. 4, given a goal and related conditions, AIPS first generates a list of subgoals by applying a set of theorems at each iteration. A value neural network is then used to evaluate these newly generated subgoals along with the previous unresolved subgoals. The top-value subgoal is selected for the next step of reasoning. This iterative process continues until the proof is successfully completed, as shown in Fig. 3(b).

### 3.3.1 Searching Proofs by Combining Value Network with Symbolic Prover

The procedure of searching for inequality proofs is generally divided into three parts: mixed reasoning for subgoal generation, evaluation, and planning.

**Subgoal Generation.** There are two methods for generating subgoals in AIPS. The first method involves applying fundamental inequality theorems. Let $X$ be the set of variables. Suppose the inequality theorem to prove is $u(X) \leq v(X)$ under a condition set $\mathcal{P}$. AIPS first homogenizes the inequality to $f(X) \leq g(X)$ on both sides by applying conditions in $\mathcal{P}$. Then, by applying theorems

---

**Algorithm 1** Generating Theorems

---

1: **function** Generate_Theorems(*expression P*, *loops N*)
2:     Initialize *Theorem Set S*,
    *Inequality Transformation Rules O, Inequality Sets A1, A2, A3*
3:     Apply $S$ to $P$ to obtain a series of inequalities and add those whose equality conditions hold to a set $R$
4:     **for** $i \leftarrow 1$ to $N$ **do**
5:         **for** each inequality $ineq$ in $R$ **do**
6:             Apply rules $O$ to $ineq$ to obtain $A1$
7:         **end for**
8:         **for** each inequality $ineq$ in $R$ **do**
9:             Apply theorems $S$ to one side of $ineq$ and check if it can be linked to the original inequality. If so, add it to $A2$
10:        **end for**
11:        **for** each inequality $ineq$ in $A2$ **do**
12:            Check if $ineq$ meets the equality condition and add it to $A3$ if it does
13:        **end for**
14:        Update $R$ by selecting $M$ inequalities from the union of $A3$ and $A1$ according to the length of inequalities
15:     **end for**
16:     **return** $R$
17: **end function**

---

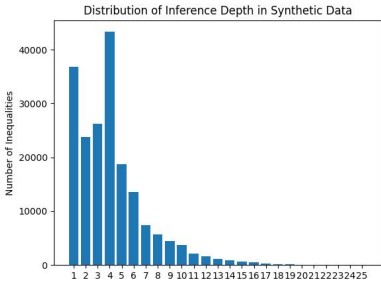
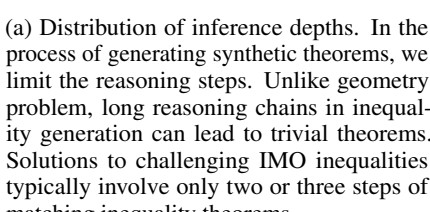

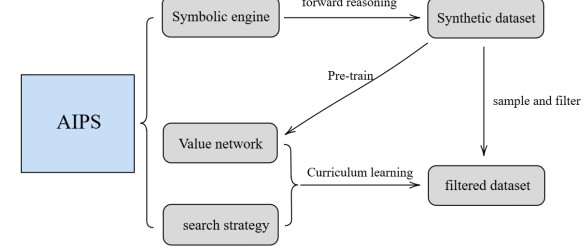

(a) Distribution of inference depths. In the process of generating synthetic theorems, we limit the reasoning steps. Unlike geometry problem, long reasoning chains in inequality generation can lead to trivial theorems. Solutions to challenging IMO inequalities typically involve only two or three steps of matching inequality theorems.

(b) Self-evolving process of AIPS. After pre-training on the initial synthetic dataset, AIPS is capable of proving some challenging theorems. Guided by the value network, it then attempts to solve problems in an increasingly difficult filtered dataset. By extracting nodes on the proof path as positive labels and other nodes as negative labels, it fine-tunes the value network and gradually improves proving performance in a curriculum manner.

Figure 3: (a) Distribution of inference depths in our dataset. (b) Self-evolving process of AIPS.

to the left-hand side of the target inequality, AIPS generates a series of new inequalities:

$$f(X) \leq h_1(X), \dots, f(X) \leq h_n(X)$$

This results in subgoals $h_i(X) \leq g(X)$. Similarly, by applying theorems to the right-hand side, AIPS also generates subgoals $f(X) \leq s_j(X)$. The second method involves applying transformation rules such as `sympy.expand` and `sympy.apart` to the goal, generating subgoals that are equivalent to the original inequality.

**Evaluation.** AIPS employs a value function $V_\theta$ to assess the difficulty of each inequality. Formally, we have a function $f$ parameterized by $\eta$ that encodes the inequality expression $s$. The encoded embedding vector $f_\eta(s)$ is then fed into a deep neural network $g_\phi$, which outputs a value in the interval [0,1]. We choose $f$ to be a transformer encoder with average pooling (Vaswani et al. 2017).

**Planning.** With the evaluation function $V_\theta$, we use the Best-First search algorithm for planning. We also test the performance of MCTS algorithm, where the result is less satisfactory.

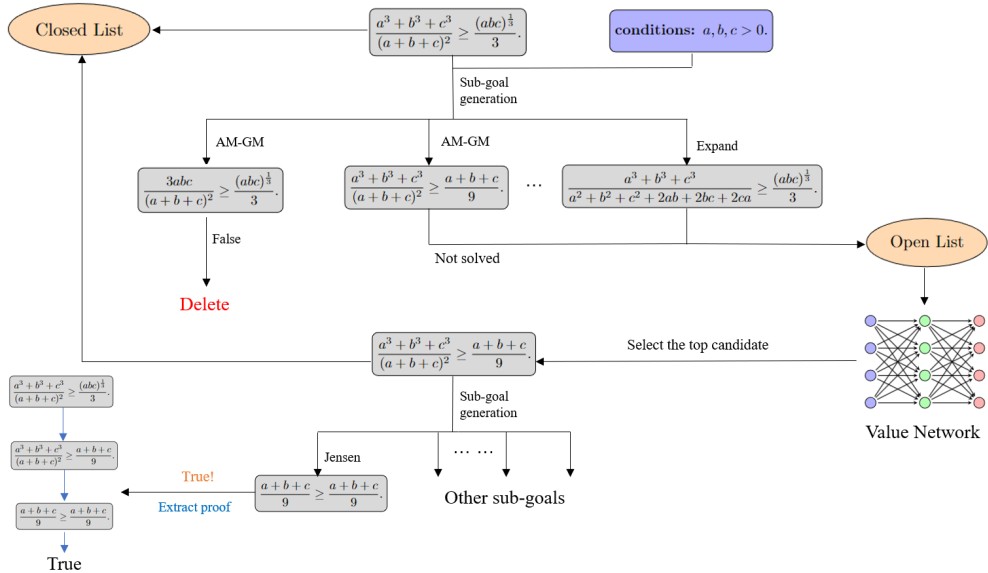

Figure 4: Overview of how AIPS proves a simple theorem. At each step, the deductive engine attempts to match inequality theorems with each side of the goal and applies all transformation rules to the expression, resulting in a list of new subgoals. The searched goal is placed into a closed list, ensuring that it will not be examined again. If one of the new subgoals is true, indicating that the inequality holds, then the theorem is proved. Otherwise, the new subgoals are added to the open list, along with other subgoals generated previously. A value network then evaluates all subgoals in the open list, and the top-value one is chosen for the next iteration of proof search.

There are two primary reasons for this. First, the action space for each state is extremely large, leading to explosive growth of the MCTS searching tree. Second, the high cost of reasoning steps makes the simulation step in MCTS nearly impractical, often exceeding time limits.

We also note that our prover can be combined with any heuristic function, and thus design various baselines in our experiments.

### 3.3.2 Pre-training Value Network Using a Heuristic Function

We define the tree-depth $\mathcal{D}$ of an inequality as the maximum depth of the expression trees on both sides. Proving an algebraic inequality is equivalent to reducing the tree-depth of the inequality to one. We use $\mathcal{D}$ as the supervision information to train initial heuristic function $f_{\text{init}}$ in the Best-First search algorithm. That is to say, we pre-train a value network $V_\theta$ as $f_{\text{init}}$ on the synthetic dataset by utilizing the tree-depth $\mathcal{D}$.

### 3.3.3 Fine-tuning Value Network on Filtered Synthetic Data

We create a new dataset by removing all inequalities with inference depth less than 4. We then randomly sample 1,200 problems and sort them by tree-depth in ascending order. For inequalities with the same tree-depth, they are sorted by the length of their string representation, with shorter lengths placed first.

The fine-tuning procedure involves sequentially proving these inequalities and updating the parameters of the value network. If an inequality is successfully proved, we record the set of subgoals on the proof path as $T$ and the set of subgoals that are searched but not on the proof path as $F$. The values of the elements in $T$ are scaled down by a factor of $\epsilon$, while the values of the elements in $F$ are increased. Using these labels, we perform a training round on the value network $V_\theta$, and then proceed to the next problem. This iterative process is used to adjust the network parameters. See Appendix **??** for more details.

# 4 Experiments

We evaluate AIPS on an Olympiad-level algebraic inequality problem test set. It outperforms the state-of-the-art methods in terms of the number of solved problems, demonstrating the strong algebraic intuitions developed by the learned value network.

## 4.1 An Olympiad-Level Inequality Benchmark

Current benchmarks for Olympiad-level math problems, such as miniF2F (Zheng et al. 2021) and Fimo (Liu et al. 2023), cover a wide array of topics but often lack a dedicated section for algebraic inequalities. In inequality benchmarks like INT (Wu et al. 2020), the problems are typically of limited difficulty. To address this gap, we collect all ternary and quaternary algebraic inequality problems from IMO since 1990. Additionally, we include challenging problems from IMO shortlists and various national mathematical Olympiads, such as the USAMO, the USA National Team Selection Tests, and the Polish, Japanese, and Korean Mathematical Olympiads, all of which are of comparable difficulty to the IMO. In total, we compile 20 problems for our test set, naming it MO-INT-20 (Math-Olympiad-INequality-Test-20). All problems are checked to ensure they are not in AIPS's training datasets. We also translate the test problems into Lean for subsequent experiments.

## 4.2 Comparison Methods

Current theorem provers include interactive theorem provers, large language models capable of generating natural language proofs, and neural symbolic theorem provers. We compare LeanCopilot (Song et al. 2024), the open-source state-of-the-art interactive theorem prover in Lean. Additionally, we evaluate general large language models like GPT-4, GPT-4 Turbo and Gemini 1.5 Pro, as well as the math-specific language model Llemma-7b (Azerbayev et al. 2023). For neural symbolic theorem provers, we design various baselines, including our deductive engine paired with breadth-first search and MCTS, our deductive engine equipped with tree-depth in Section 3.3.2 or LLM heuristics as the value function, and our AIPS with only pretrained value network.

It should be noted that we cannot compare with several existing interactive theorem provers (Polu and Sutskever 2020; Polu et al. 2022) since these provers are not open source to be reproduced. However, it is reported that these provers can only prove a few early Olympiad inequalities, as detailed in the appendix of their respective papers.

## 4.3 Comparison Results and Analysis

We test 11 different provers on the inequalities in MO-INT-20, with each problem limited to 90 minutes of solving time, consistent with the standard problem-solving time in the IMO. All neural-symbolic provers are tested on a single CPU core (equivalent to 1.5 CPU hours per problem). The comparison results are shown in Table 1. It can be seen that our AIPS achieves the best performance and solves 10 out of 20 problems.

Table 1: Model Performances on the MO-INT-20. **DE denotes our deductive engine**. BFS and MCTS are Breadth-First Search and Monte Carlo Tree Search, respectively.

| Model Category | Model | Problems Solved (20) |
|---|---|---|
| Large Language Models | Gemini 1.5 Pro | 1 |
| | GPT-4 | 0 |
| | GPT-4 Turbo | 0 |
| | Llemma-7b | 0 |
| Interactive Theorem Provers | LeanCopilot (LeanDojo) | 0 |
| Neural-Symbolic Provers | DE + GPT-4 Turbo's heuristics | 6 |
| | DE + BFS | 4 |
| | DE + MCTS | 5 |
| | DE + tree-depth heuristic function | 7 |
| | AIPS with pretrained value network | 7 |
| | AIPS | 10 |

**Analysis of Large Language Models' Performance.** Large language models like GPT-4 have demonstrated remarkable reasoning abilities (Lewkowycz et al. 2022; Wei et al. 2022). However, in this test, only one of the four models, Gemini 1.5 Pro, successfully generates a fully correct natural language proof. When solving problems, large language models tend to either make trivial mistakes or indicate that they do not know how to solve them, despite the potential contamination of their training data by online proofs. These results reveal their limited math reasoning ability.

**Analysis on a Formal Theorem Prover's Performance.** Recent studies reveal the capabilities of neural theorem provers based on Interactive Theorem Prover (ITP) frameworks (Yang et al. 2024; Rute et al. 2024). These systems generally convert theorem proving into code completion tasks. We evaluate the performance of one such theorem prover, LeanCopilot (Song et al. 2024), developed from LeanDojo, on our test set. LeanCopilot is the current open-source state-of-the-art theorem prover based on Lean. The results indicate its limited ability to solve complex algebraic problems: None of the problems are solved through proof search in LeanCopilot. Additional tests on tactic suggestions (see Appendix **??**) show that current formal theorem provers struggle to predict the complex premises required for proving inequalities.

**Analysis on Neural Symbolic Provers' Performance.** In this test, neural symbolic provers demonstrate a strong ability to prove algebraic inequalities using best-first search algorithm. By applying either breadth-first search or MCTS algorithm, our deductive engine successfully solves four and five problems, respectively. We also test performance under the guidance of a tree-depth heuristic function and a pre-trained value network using the best-first search algorithm, both of which solve seven problems. Additionally, we prompt GPT-4 Turbo and find it exhibit some algebraic intuition, successfully guiding the deductive engine to solve six problems—two more than the breadth-first search. However, it is worth noting that large language models (LLMs) may occasionally prioritize lengthy and meaningless subgoals. Due to the exponential growth of the number of new inequalities as the width and height of the expression trees increase, it can result in expression strings longer than the LLMs' input context length. For example in problem 4 from the 2014 Japan Mathematical Olympiad, it chooses a very long subgoal at iteration 2, resulting in subgoals at the next iteration being three times longer than its input context length.

Finally, following a curriculum learning strategy on 1,000 inequality problems, AIPS achieves the best performance, solving 10 out of 20 problems. Among the 10 problems from the IMO or IMO shortlist, it successfully solves five, reaching the average level of IMO contestants. We also test the performances of AIPS after 200, 400, 600, and 800 loops of fine-tuning value network (see Appendix **??**). The results demonstrate that our value curriculum learning strategy is very effective, with the number of proof search steps significantly decreasing during the training process, and the number of solved problems increasing to 10 ultimately.

# 5  Conclusion

In conclusion, solving Olympiad-level mathematical problems is a significant milestone in machine intelligence and automated reasoning. The lack of large-scale, high-quality datasets presents a challenge, particularly in algebraic systems. To address this, we propose *AIPS*, an *Algebraic Inequality Proving System*, which autonomously generates complex inequality theorems and effectively solves Olympiad-level inequality problems without human input. Utilizing a value curriculum learning strategy, AIPS demonstrated strong mathematical intuition by solving 10 out of 20 International Mathematical Olympiad-level problems. One of these theorems was selected for a major city's 2024 Mathematical Olympiad.

In the future, by incorporating more fundamental theorems and operational rules, our AIPS could solve even more complex problems, discover a greater number of non-trivial theorems, and assist mathematicians in solving modern mathematical challenges. However, it currently lacks the ability to autonomously propose and comprehend new definitions. Instead, it relies on handwritten theorems and matching rules, which is time-consuming. Addressing this limitation is a crucial area for future research.

# 6 Acknowledgements

We extend our heartfelt gratitude to the three distinguished contestants—two National Mathematical Olympiad gold medalists and one silver medalist—for their invaluable evaluations of our synthetic theorems. We also express our sincere thanks to their coach Zhibin Liang, whose efforts made this collaboration possible. Furthermore, we deeply appreciate the insightful discussions from Jiajun Song, Yuxuan Wang, and Dr. Chi Zhang at Beijing Institute for General Artificial Intelligence. This work was supported in part by the National Natural Science Foundation of China under Grants 61976214.

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
