# Appendix

## 1 Technical Details of the Deductive Engine

We provide more information on AIPS' deductive engine and the training process for the value network. To highlight the reasoning ability and maintain readability of proofs, we avoid using brute-force methods such as augmentation-substitution and Wu's method Wu [1978].

### 1.1 Background

#### 1.1.1 Basic Knowledge in Theorem Proving

Theorem proving encompasses two types of reasoning: forward reasoning and backward reasoning. Forward reasoning involves identifying a pattern match between a particular theorem and the given conditions along with the universal variables, then deducing the conclusion. In contrast, backward reasoning works in the opposite direction, where the conclusion and variables are matched with a specific theorem, breaking down the main goal into smaller, more manageable subgoals. Both

38th Conference on Neural Information Processing Systems (NeurIPS 2024) Track on Datasets and Benchmarks.

methods are essential in constructing and navigating the logical steps to establish the validity of complex mathematical theorems, as shown in Fig. 1.

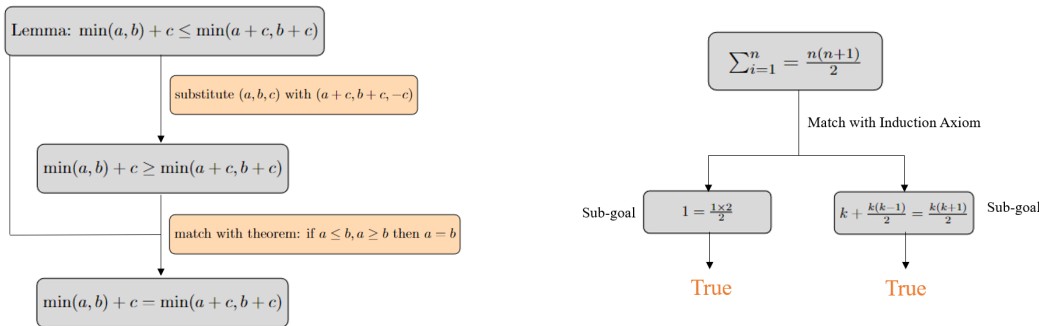

Figure 1: Two examples of forward reasoning on the left and backward reasoning on the right.

### 1.1.2 Challenges in Algebraic Reasoning

There are two main challenges in reasoning within algebraic systems. The first is the infinite reasoning space within finite conditions, caused by the numerous possible expression trees and the vast search space for premises. This contrasts with solving Euclidean geometry problems, where a deduction fixed point exists with respect to a set of geometric rules or axioms. To address this issue, we consider only the current expression tree at each step of reasoning. The second challenge lies in pattern matching, which requires accurately identifying and applying relevant theorems to given sub-structures. For theorems with function-type variables, like Jensen's Inequality, pattern matching is more challenging and time-consuming. We provide heuristic functions to identify possible structures where Jensen's Inequality can be applied.

## 1.2 Theorems, Rules and Pattern Matching

### 1.2.1 Theorems, Methods and Transformation Rules

Our deductive engine incorporates six well-known inequality theorems frequently used in mathematical Olympiads, several one-variable inequality scaling and solving methods, and dozens of algebraic transformation rules. The inequality theorems include the **Arithmetic Mean-Geometric Mean (AM-GM) inequality**, the **weighted AM-GM inequality**, **Hölder's inequality**, **Jensen's inequality**, **Schur's inequality**, and **Müirhead's theorem**. For simplicity, we have excluded some theorems that can be directly proved using these inequalities, such as the Geometric Mean-Harmonic Mean (GM-HM) inequality and the Cauchy-Schwarz inequality.

Here we list some frequently used methods and transformation rules:

- `nodiv_expr`: Multiply both sides to eliminate denominators
- `nomul_expr`: Divide both sides by all factors
- `no_sep_denom`: Combine fractions on both sides
- `sep_neg`: Move terms with negative coefficients to the other side of the inequality
- `zero_side`: Subtract one side from the other to make one side equal to zero
- `no_pow`: Remove roots at the second level from the top of the expression tree on both sides
- `try_together_l`, `try_together_r`: Combine fractions on the left or right side
- `try_expand_l`, `try_expand_r`: Expand expressions on the left or right side
- `all_cyc_mul_expr`: Multiply both sides by a cyclically symmetric polynomial, with one of its generators on either the left or right side of the inequality (a generator is a term that, when cyclically permuted, generates the expression)
- `try_factor_both`: Factorize both sides
- `check_one_var`: Check if the solution of a one-variable inequality is contained in a given interval
- `check_linear_ctr`: Check if a one-variable expression can be applied with tangent line trick
- `find_main_fun`: For a cyclically symmetric expression, try to find a function that can match with Jensen's inequality as well as generate this expression

### 1.2.2 Pattern Matching

An important step in generating synthetic theorems is matching algebraic expressions with these theorems. We use the AM-GM inequality as an example to illustrate pattern matching method as follows.

**Theorem 1.** *(AM-GM) For non-negative real numbers $a_1, a_2, \ldots, a_n$,*

$$a_1 + a_2 + \cdots + a_n \geq n \sqrt[n]{a_1 a_2 \cdots a_n}$$

*with equality if and only if $a_1 = a_2 = \cdots = a_n$.*

Assuming all variables are non-negative, pattern matching for an algebraic expression with the AM-GM inequality (on the Left-Hand-Side) is explained in three steps:

1. Traverse through the expression tree, and label a node with $1$ if the whole expression value increases as the value of the node increases, with $-1$ if the expression value decreases as the value of the node increases, and with *None* if this cannot be determined.

2. At each node labeled $1$ or $-1$ and calculated with an *Add* operation, find all non-negative sub-arguments of the node's expression and place them in nonneg_set. Similarly, find all non-positive sub-arguments and place them in nonpos_set.

3. For the obtained sets nonneg_set and nonpos_set, we use the following method to match the mean inequalities:

   - Arbitrarily partition each set into multiple subsets.

- the sum of the elements in each subset can be used as a variable to match the left side of the mean inequality.
- If a subset does not contribute to the inequality, it is excluded from the partition.

This process allows us to identify all possible mean inequalities that can be matched. We then replace the original sub-expressions in the expression tree with the transformed ones based on the matched inequalities. By doing so, a new inequality is derived according to the labels.

## 1.3 Details of Synthetic Data Generation

Olympiad inequalities aim for not only difficulty but also conciseness and elegance, a principle also valued in modern mathematics. Although our deductive engine can generate various types of inequalities, we focus on cyclically symmetric inequalities in semi-definite systems that can be generated with a limited number of steps to avoid lengthy and chaotic expressions.

Initially, we generate thousands of premises as the starting points for data generation using Algorithm 1. For each generated premise, we run the data-generation algorithm described in the main paper. During this process, we discard inequalities for which equality does not hold or which do not have the desired form, and halt the generation after a maximum of 25 iterations of search. Utilizing 32 CPUs over an 8-hour period, the deductive engine produces 191,643 theorems. This demonstrates the engine's ability to efficiently generate a large number of high-quality inequality theorems, thereby addressing the bottleneck of lacking a high-quality dataset for learning-based provers.

---

**Algorithm 1** Generating Initial Premises

---

**function** GENERATE_EXPRESSIONS(*variable_list I*, *loop_limit N*)
   Initialize *Results* and *Basic_Operations*
   **for** $i \leftarrow 1$ to $N$ **do**
     Initialize *New_Expressions*
     **for** each pair $(a, b)$ in $I$ and each operation $f$ in *Basic_Operations* **do**
       Add $f(a, b)$ to *New_Expressions*
     **end for**
     Add *New_Expressions* to $I$
   **end for**
   **for** each expression *expr* in $I$ **do**
     Add cyclic summation of *expr* to *Results*
   **end for**
   **return** *Results*
**end function**

---

## 2 Experiments and Analysis

In this section, we provide details of our experiments and present the test results. We also include technical analysis of these results.

### 2.1 Synthetic Dataset Statistics

We conduct a statistical analysis on the synthetic dataset, focusing on inequality lengths (in string representation) and *tree-depth* (the maximum expression tree height on both sides of an inequality), as depicted in Figure 2. The distributions of lengths and tree-depth are related to the difficulty and search complexity. These distributions illustrate that our theorems range from simple to complex, reflecting a spectrum of difficulty levels in our dataset.

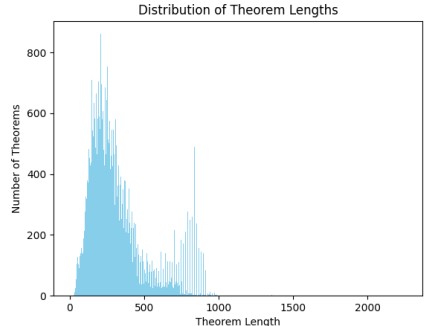
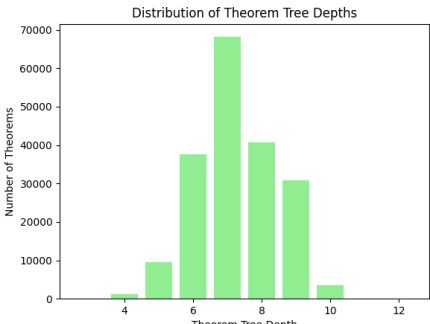

Figure 2: Ditribution of lengths and tree-depths of synthetic theorems.

### 2.2 Details of Value Curriculum Learning

The value network $V_\theta$ comprises two main components: the pre-trained transformer encoder, Llemma-7b [Azerbayev et al., 2023], followed by a $4096 \times 256 \times 1$ multilayer perceptron that outputs a value in the interval $(0, 1)$. Initially, AIPS successfully resolves 7 out of 20 problems from the test set using the pre-trained value network.

The value network $V_\theta$ functions as the heuristic in the best-first-search algorithm. It comprises two main components: the pre-trained transformer encoder, Llemma-7b, followed by a $4096 \times 256 \times 1$ feedforward neural network that outputs a value in the interval $(0, 1)$. Initially, AIPS successfully resolves 7 out of 20 problems from the test set using the pre-trained value network.

The procedure of value curriculum learning is as follows. After successfully proving a theorem, each node along the proof path is relabeled with a value that is $\epsilon$ times its original value. For node that has been searched but is not part of the proof path, if its original label is $v$, the label of this node is updated at the end of this curriculum learning round according to the formula: $\max(m, v) \times \eta + 1 - \eta$. Here $m$ represents the maximum value after modification among the proof path nodes. Subsequently, the relabeled nodes undergo 10 loops of fine-tuning training. We choose $\epsilon = 0.3$ and $\eta = 0.7$.

Before the value curriculum learning process, we randomly select 1,200 theorems from the synthetic dataset, excluding theorems with an inference depth of less than 4. These theorems undergo a curriculum learning strategy tailored for the pre-trained model. We limit the time for solving each problem to 40 minites. During curriculum learning, the theorems are solved and trained in an ascending order, sorted first by tree-depth, then by theorem length. The first 150 problems are solved within a mere two hours. After four days of training, AIPS solves 892 out of the first 1,000 problems, with 887 successes in the first 950 theorems. Since it struggles to solve problems after the 950th theorem, we decide to halt the training process at the 1,000th problem.

### 2.3 Performance Analysis During Curriculum Learning

The extensive experiments verify that the value curriculum learning strategy is very effective. The number of search loops required to solve testing theorems decreases noticeably throughout the train-

ing process, enabling AIPS to successfully solve 10 out of 20 IMO-level inequality problems using an RTX-4090 GPU and a single CPU. Fig. 3 shows the decreasing number of search loops during curriculum learning on the 2001 IMO Problem 2, and Fig. 4 shows the increasing number of solved problems during curriculum learning.

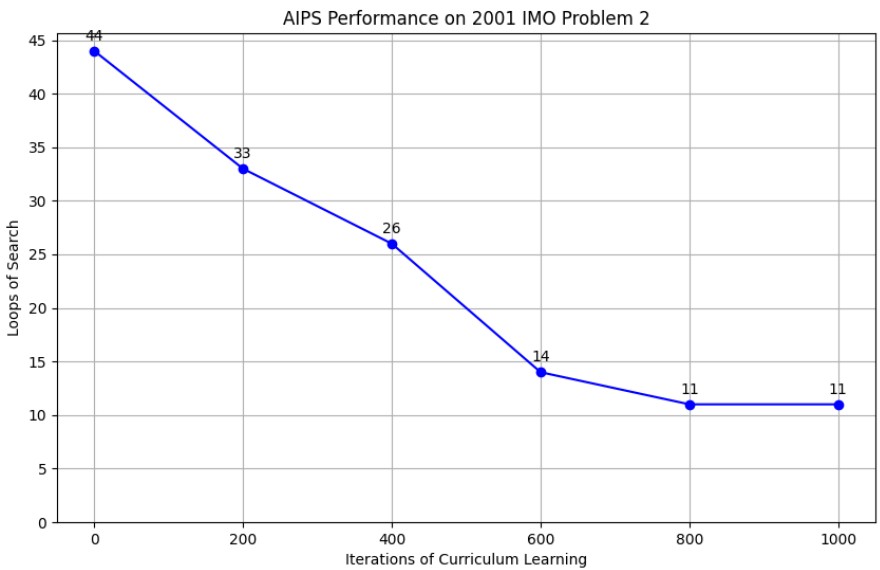

Figure 3: AIPS progressively finds the proof path more efficiently throughout the training process.

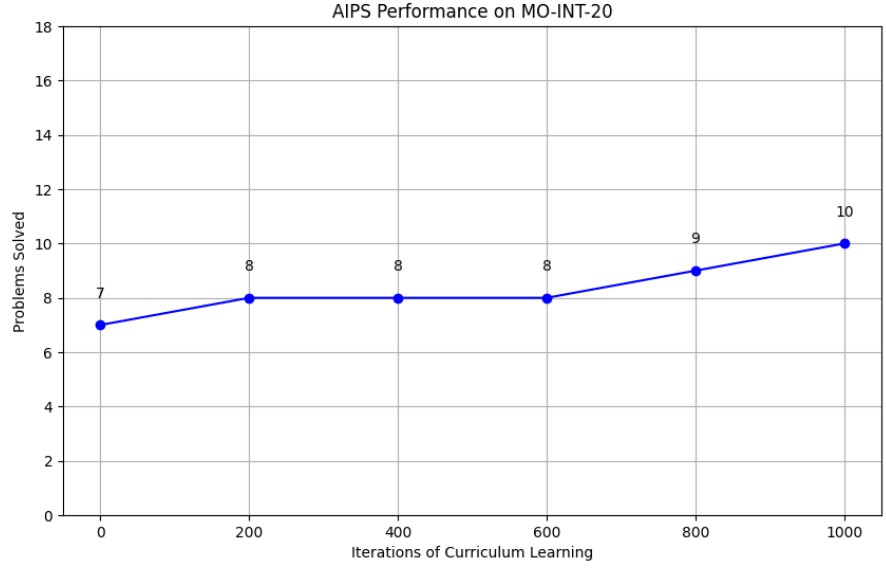

Figure 4: AIPS solves more problems with the increasing iterations of value curriculum learning.

## 2.4 Our Benchmark: Mathematical-Olympiad-INequality-Test-20

We collect all ternary and quaternary algebraic inequality problems from IMO since 1990, some challenging problems from IMO shortlists and several national mathematical Olympiads, such as

the USAMO, the USA National Team Selection Tests, the Polish/Korean/Japanese Mathematical Olympiad, all of which are of comparable difficulty to the IMO. The collected 20 problems provide a new challenging benchmark for the realm of automatic theorem proving, dubbed as MO-INT-20 (Math-Olympiad-INequality-Test-20). The details of these 20 problems are as follows.

- **Problem 1 (IMO 1990 Shortlist):**
  For $a > 0, b > 0, c > 0, d > 0$ such that $a \cdot b + b \cdot c + c \cdot d + d \cdot a = 1$, show that:
  $$\frac{a^3}{b+c+d} + \frac{b^3}{c+d+a} + \frac{c^3}{d+a+b} + \frac{d^3}{a+b+c} \geq \frac{1}{3}$$

- **Problem 2 (IMO 1993 Shortlist):**
  For $a > 0, b > 0, c > 0, d > 0$, show that:
  $$\frac{a}{b+2c+3d} + \frac{b}{3a+c+2d} + \frac{c}{2a+3b+d} + \frac{d}{a+2b+3c} \geq \frac{2}{3}$$

- **Problem 3 (IMO 1995 P2):**
  For $a > 0, b > 0, c > 0$ such that $a \cdot b \cdot c = 1$, show that:
  $$\frac{1}{c^3(a+b)} + \frac{1}{b^3(a+c)} + \frac{1}{a^3(b+c)} \geq \frac{3}{2}$$

- **Problem 4 (IMO 1996 Shortlist):**
  For $a > 0, b > 0, c > 0$ such that $a \cdot b \cdot c = 1$, show that:
  $$\frac{a \cdot b}{a^5 + a \cdot b + b^5} + \frac{a \cdot c}{a^5 + a \cdot c + c^5} + \frac{b \cdot c}{b^5 + b \cdot c + c^5} \leq 1$$

- **Problem 5 (USAMO 1997 P5):**
  For $a > 0, b > 0, c > 0$, show that:
  $$\frac{1}{a^3 + b^3 + a \cdot b \cdot c} + \frac{1}{b^3 + c^3 + a \cdot b \cdot c} + \frac{1}{c^3 + a^3 + a \cdot b \cdot c} \leq \frac{1}{a \cdot b \cdot c}$$

- **Problem 6 (IMO 1998 Shortlist A3):**
  For $a > 0, b > 0, c > 0$ such that $a \cdot b \cdot c = 1$, show that:
  $$\frac{a^3}{(1+b)(1+c)} + \frac{b^3}{(1+c)(1+a)} + \frac{c^3}{(1+a)(1+b)} \geq \frac{3}{4}$$

- **Problem 7 (IMO 2000 P2):**
  For $a > 0, b > 0, c > 0$ such that $a \cdot b \cdot c = 1$, show that:
  $$(a - 1 + \frac{1}{b})(b - 1 + \frac{1}{c})(c - 1 + \frac{1}{a}) \leq 1$$

- **Problem 8 (IMO 2001 P2):**
  For $a > 0, b > 0, c > 0$, show that:
  $$\frac{a}{\sqrt{a^2 + 8bc}} + \frac{b}{\sqrt{8ac + b^2}} + \frac{c}{\sqrt{8ab + c^2}} \geq 1$$

- **Problem 9 (USAMO 2003 P5):**
  For $a > 0, b > 0, c > 0$, show that:
  $$\frac{(a + b + 2c)^2}{2c^2 + (a+b)^2} + \frac{(a + 2b + c)^2}{2b^2 + (a+c)^2} + \frac{(2a + b + c)^2}{2a^2 + (b+c)^2} \leq 8$$

- **Problem 10 (Poland 2004):**
  For $a > 0, b > 0, c > 0, d > 0$, show that:
  $$\frac{a}{(a^3 + 63bcd)^{\frac{1}{3}}} + \frac{b}{(63acd + b^3)^{\frac{1}{3}}} + \frac{c}{(63abd + c^3)^{\frac{1}{3}}} + \frac{d}{(63abc + d^3)^{\frac{1}{3}}} \geq 1$$

- **Problem 11 (IMO 2004 Shortlist A5):**
  For $a > 0, b > 0, c > 0$ such that $a \cdot b + b \cdot c + c \cdot a = 1$, show that:

$$\left(\frac{1}{a} + 6b\right)^{\frac{1}{3}} + \left(\frac{1}{b} + 6c\right)^{\frac{1}{3}} + \left(\frac{1}{c} + 6a\right)^{\frac{1}{3}} \le \frac{1}{a \cdot b \cdot c}$$

- **Problem 12 (IMO 2006 P3):**
  Given real numbers $a, b, c$, show that:

$$|ab(a^2 - b^2) + bc(b^2 - c^2) + ca(c^2 - a^2))| \le \frac{9}{16\sqrt{2}}(a^2 + b^2 + c^2)^2$$

- **Problem 13 (IMO 2009 Shortlist):**
  For $a > 0, b > 0, c > 0$ such that $\frac{1}{a} + \frac{1}{b} + \frac{1}{c} = a + b + c$, show that:

$$(2a + b + c)^{-2} + (a + 2b + c)^{-2} + (a + b + 2c)^{-2} \le \frac{3}{16}$$

- **Problem 14 (USA IMO Team Selection 2010 P2):**
  For $a > 0, b > 0, c > 0$ such that $a \cdot b \cdot c = 1$, show that:

$$\frac{1}{c^5(a + 2b)^2} + \frac{1}{b^5(2a + c)^2} + \frac{1}{a^5(b + 2c)^2} \ge \frac{1}{3}$$

- **Problem 15 (USAMO 2011 P1):**
  For $a > 0, b > 0, c > 0$ such that $a^2 + b^2 + c^2 + (a + b + c)^2 \le 4$, show that:

$$\frac{a \cdot b + 1}{(a + b)^2} + \frac{b \cdot c + 1}{(b + c)^2} + \frac{c \cdot a + 1}{(c + a)^2} \ge 3$$

- **Problem 16 (Korea 2011 P4):**
  For $a \ge 0, b \ge 0, c \ge 0$ such that $a + b + c = 1$, show that:

$$\frac{1}{a^2 - 4a + 9} + \frac{1}{b^2 - 4b + 9} + \frac{1}{c^2 - 4c + 9} \le \frac{7}{18}$$

- **Problem 17 (USAMO 2012):**
  For $a > 0, b > 0, c > 0$, show that:

$$\frac{b^3 + 3c^3}{5b + c} + \frac{a^3 + 3b^3}{5a + b} + \frac{3a^3 + c^3}{a + 5c} \ge \frac{2}{3}(a^2 + b^2 + c^2)$$

- **Problem 18 (Japan 2014 P5):**
  For $a \ge 0, b \ge 0, c \ge 0$ such that $a + b + c = 1$, show that:

$$\frac{a}{9bc + 4(b - c)^2 + 1} + \frac{b}{9ac + 4(-a + c)^2 + 1} + \frac{c}{9ab + 4(a - b)^2 + 1} \ge \frac{1}{2}$$

- **Problem 19 (USAMO 2017 P6):**
  For $a \ge 0, b \ge 0, c \ge 0, d \ge 0$ such that $a + b + c + d = 4$, show that:

$$\frac{a}{b^3 + 4} + \frac{b}{c^3 + 4} + \frac{c}{d^3 + 4} + \frac{d}{a^3 + 4} \ge \frac{2}{3}$$

- **Problem 20 (IMO 2020 P2):**
  For $a \ge b, b \ge c, c \ge d, d > 0$ such that $a + b + c + d = 1$, show that:

$$(a + 2b + 3c + 4d)a^a b^b c^c d^d < 1$$

## 2.5 Details of Comparison Methods and Testing Results

### 2.5.1 Monte-Carlo Tree Search

We evaluate the performance of Monte-Carlo Tree Search (MCTS). Compared to games like Go or chess, theorem proving can have an extremely large or even infinite action space, since applying each theorem or axiom usually comes with a set of parameters. Therefore, a direct application of MCTS to our problems is infeasible. To address this, we need to modify the MCTS algorithm.

First, we place a restriction on our action space: at each state, we sample all possible actions generated from the current proof state, then sort them according to a tree-depth heuristic function, which evaluates the difficulty of the proof state after applying them, and pick the first $k$ proof states (we choose $k = 5$). During the selection step in MCTS, we apply the Upper Confidence Bounds algorithm,

$$\text{SelectedAction} = \text{Argmax}_i \left( v_i + C \cdot \sqrt{\frac{\ln(N)}{n_i}} \right)$$

Here, $n_i$ is the number of visits of the $i$-th action (proof state), $N$ is the number of visits of the current state, $C$ is an adjustable hyperparameter (we choose $C = 0.3\sqrt{2}$), and $v_i$ is the value of the state after applying the $i$-th action.

We use two methods to compute $v_i$: either it is directly given by the heuristic evaluation function, or we execute a best-first search to a constant depth $d$ ($d = 2$), and take the value of the current best state as $v_i$.

### 2.5.2 Testing Examples from Large Language Models

Large Language Models (LLMs) currently struggle to solve IMO-level inequalities through the generation of natural language proofs. In our evaluation of 4 LLMs—GPT-4, GPT-4 Turbo, Gemini 1.5 Pro, and Llemma-7b—only Gemini 1.5 Pro successfully generates a correct solution to USAMO-1997-P5. Proofs generated by GPT-4 and GPT-4 Turbo contain some logical errors, while Llemma-7b rarely produces meaningful reasoning sentences. We present the results of first three LLMs below, including the only correct proof by Gemini 1.5 Pro to USAMO-1997-P5, and the incorrect proofs by using these three LLMs to IMO-1995-P2.

**Statement**: (USAMO 1997 P5) Given three positive real numbers $a, b, c$, show that

$$\frac{1}{a^3 + b^3 + abc} + \frac{1}{b^3 + c^3 + abc} + \frac{1}{c^3 + a^3 + abc} \le \frac{1}{abc}$$

---

**Correct proof**:

1. Factorization: Recall the factorization of the sum of cubes:

$$x^3 + y^3 = (x + y)(x^2 - xy + y^2)$$

2. Apply AM-HM: For non-negative numbers $x$ and $y$, the AM-HM inequality states:

$$\frac{x + y}{2} \ge \frac{2}{1/x + 1/y}$$

Rearranging the inequality:

$$x^2 + y^2 - xy \ge xy$$

3. Manipulate the given inequality: Lets focus on the first term and use the factorization:

$$\frac{1}{a^3 + b^3 + abc} = \frac{1}{(a + b)(a^2 - ab + b^2) + abc}$$

Since $a^2 - ab + b^2 \ge ab$ (from step 2), we have:

$$\frac{1}{(a + b)(a^2 - ab + b^2) + abc} \le \frac{1}{(a + b)ab + abc} = \frac{1}{(a + b + c)ab}$$

4. Apply similar reasoning to other terms: Similarly, for the second and third terms:

$$\frac{1}{b^3 + c^3 + abc} \le \frac{1}{(a + b + c)bc}$$

$$\frac{1}{c^3 + a^3 + abc} \le \frac{1}{(a + b + c)ca}$$

5. Combine the inequalities: Adding the three inequalities we just derived and simplify the right hand side:

$$\frac{1}{a^3 + b^3 + abc} + \frac{1}{b^3 + c^3 + abc} + \frac{1}{c^3 + a^3 + abc} \le \frac{1}{abc}$$

This completes the proof.

Figure 5: Correct proof generated by Gemini 1.5 Pro.

**Statement**: (IMO 1995 P2) Given positive real numbers $a, b, c$ such that $abc = 1$, show that

$$\frac{1}{c^3(a+b)} + \frac{1}{b^3(a+c)} + \frac{1}{a^3(b+c)} \geq 3/2$$

---

**Incorrect proof:** (Generated by Gemini 1.5 Pro)

Transformation: First, let's simplify the given expression by multiplying the numerator and denominator of each term by abc (which is equal to 1 due to the condition abc = 1):

$$\frac{1}{c^3(a+b)} + \frac{1}{b^3(a+c)} + \frac{1}{a^3(b+c)} = \frac{ab}{c^2(a+b)} + \frac{ac}{b^2(a+c)} + \frac{bc}{a^2(b+c)}$$

Applying AM-HM: Now, consider the following positive numbers:

$$x_1 = c^2(a+b), x_2 = b^2(a+c), x_3 = a^2(b+c)$$

Applying AM-HM inequality to these:

$$\frac{c^2(a+b) + b^2(a+c) + a^2(b+c)}{3} \geq \frac{3}{\frac{1}{c^3(a+b)} + \frac{1}{b^3(a+c)} + \frac{1}{a^3(b+c)}}$$

Simplifying: Expanding and simplifying the left side:

$$\frac{a^2b + ab^2 + ac^2 + a^2c + b^2c + bc^2}{3} \geq \frac{3}{\frac{ab}{c^2(a+b)} + \frac{ac}{b^2(a+c)} + \frac{bc}{a^2(b+c)}}$$

Since abc = 1, we can substitute and further simplify:

$$\frac{ab/c + ac/b + bc/a}{3} \geq \frac{3}{\frac{1}{c^3(a+b)} + \frac{1}{b^3(a+c)} + \frac{1}{a^3(b+c)}}$$

Final Step: Invert both sides of the inequality (remembering to flip the inequality sign):

$$\frac{1}{c^3(a+b)} + \frac{1}{b^3(a+c)} + \frac{1}{a^3(b+c)} \geq 3/2$$

This proves the required inequality.

Figure 6: Incorrect proof generated by Gemini 1.5 Pro. The errors are labeled in blue.

**Statement**: (IMO 1995 P2) Given positive real numbers $a, b, c$ such that $abc = 1$, show that

$$\frac{1}{c^3(a+b)} + \frac{1}{b^3(a+c)} + \frac{1}{a^3(b+c)} \geq 3/2$$

---

**Incorrect proof:** (Generated by ChatGPT4 Turbo)

To prove this inequality, we can apply the AM-GM (Arithmetic Mean - Geometric Mean) inequality as follows:

Given $abc = 1$:

By applying the AM-GM inequality:

$$a/b + b/a \geq 2, b/c + c/b \geq 2, c/a + a/c \geq 2$$

Summing these inequalities:

$$(a/b + b/a) + (b/c + c/b) + (c/a + a/c) \geq 6$$

Rearrange and factor:

$$a^2/b^2 + b^2/c^2 + c^2/a^2 + b^2/a^2 + c^2/b^2 + a^2/c^2 \geq 6$$

Now consider the expressions under the original inequality:

$$\frac{1}{a^3(b+c)} = \frac{1}{a^3(b+c)}$$

Since abc = 1, we can write this as:

$$= \frac{b^3 c^3}{a^3(b+c)} = \frac{(bc)^3}{a^3(b+c)}$$

By AM-GM, $b + c \geq 2\sqrt{bc}$. Therefore:

$$\frac{1}{a^3(b+c)} \geq \frac{(bc)^3}{2a^3\sqrt{bc}} = \frac{b^2 c^2}{2a^3\sqrt{bc}} \cdot \sqrt{bc} = \frac{b^{5/2} c^{5/2}}{2a^3}$$

We can apply similar transformations to the other fractions:

$$\frac{1}{b^3(c+a)} \geq \frac{a^{5/2} c^{5/2}}{2b^3}, \frac{1}{c^3(a+b)} \geq \frac{a^{5/2} b^{5/2}}{2c^3}$$

Adding these up:

$$\frac{1}{a^3(b+c)} + \frac{1}{b^3(c+a)} + \frac{1}{c^3(a+b)} \geq \frac{b^{5/2} c^{5/2}}{2a^3} + \frac{a^{5/2} c^{5/2}}{2b^3} + \frac{a^{5/2} b^{5/2}}{2c^3}$$

Using our earlier inequality

$$(a^2/b^2 + b^2/a^2 + b^2/c^2 + c^2/b^2 + c^2/a^2 + a^2/c^2 \geq 6)$$

we can apply the generalized AM-GM inequality:

$$\frac{1}{a^3(b+c)} + \frac{1}{b^3(c+a)} + \frac{1}{c^3(a+b)} \geq \frac{1}{2} \cdot (\frac{b^{5/2} c^{5/2}}{a^3} + \frac{a^{5/2} c^{5/2}}{b^3} + \frac{a^{5/2} b^{5/2}}{c^3})$$

Each of these terms is equal to 1 by our substitution using abc = 1, simplifying this expression:

$$\geq 1/2 + 1/2 + 1/2 = 3/2$$

Therefore, we have proved:

$$\frac{1}{a^3(b+c)} + \frac{1}{b^3(c+a)} + \frac{1}{c^3(a+b)} \geq 3/2$$

Figure 7: Incorrect proof generated by ChatGPT4 Turbo. The errors are labeled in blue.

**Statement**: (IMO 1995 P2) Given positive real numbers $a, b, c$ such that $abc = 1$, show that

$$\frac{1}{c^3(a+b)} + \frac{1}{b^3(a+c)} + \frac{1}{a^3(b+c)} \geq 3/2$$

**Incorrect proof:** (Generated by ChatGPT 4)

The Arithmetic Mean - Geometric Mean (AM-GM) Inequality states:

$$\frac{x_1 + x_2 + \cdots + x_n}{n} \geq \sqrt[n]{x_1 \cdot x_2 \ldots x_n}$$

with equality if and only if $x_1 = x_2 \cdots = x_n$. Let's consider using AM-GM on the denominators.
Analyzing Each Fraction
For $\frac{1}{c^3(a+b)}$: Since $abc = 1$, we have $c^3 = \frac{1}{a^3 b^3}$. Hence,

$$\frac{1}{c^3(a+b)} = \frac{a^3 b^3}{a+b}$$

Similiarly:

$$\frac{1}{b^3(a+c)} = \frac{a^3 c^3}{a+c}, \frac{1}{a^3(b+c)} = \frac{b^3 c^3}{b+c}$$

Here each fraction can be analyzed by using AM-GM on the denominators.

$$a + b \geq 2\sqrt{ab}, b + c \geq 2\sqrt{bc}, c + a \geq 2\sqrt{ca}$$

Given $abc = 1$, we also have $\sqrt{ab} = \frac{c}{\sqrt{abc}} = c$ and similarly for others. Thus,

$$a + b \geq 2c, b + c \geq 2a, c + a \geq 2b$$

Substituting back:

$$\frac{a^3 b^3}{a+b} \geq \frac{a^3 b^3}{2c}, \frac{b^3 c^3}{b+c} \geq \frac{b^3 c^3}{2a}, \frac{c^3 a^3}{c+a} \geq \frac{c^3 a^3}{2b}$$

Since $a^3 b^3 = c^3, b^3 c^3 = a^3, c^3 a^3 = b^3$ from $abc = 1$,

$$\frac{a^3 b^3}{2c} = \frac{c^2}{2}, \frac{b^3 c^3}{2a} = \frac{a^2}{2}, \frac{c^3 a^3}{2b} = \frac{b^2}{2}$$

Adding these,

$$\frac{a^2}{2} + \frac{b^2}{2} + \frac{c^2}{2} = \frac{a^2 + b^2 + c^2}{2}$$

To conclude, we need the final step:

$$a^2 + b^2 + c^2 \geq 3$$

which holds by AM-GM as $a^2 + b^2 + c^2 \geq 3\sqrt[3]{a^2 b^2 c^2} = 3$. This completes the proof.

Figure 8: Incorrect proof generated by ChatGPT 4. The errors are labeled in blue.

### 2.5.3   Testing Examples from LeanCopilot

We also test LeanCopilot (developed from LeanDojo Yang et al. [2024]), the current open-source state-of-the-art theorem prover based on Lean. It solves none of the 20 problems, all resulting in time-limited feedback. We present some results of the tactic search provided by LeanCopilot in Figure 9. It can be seen that the prover tends to rely heavily on built-in tactics and struggles to predict complex premises.

```
USAMO−1997−P5 :

theorem (a b c:R)(h0:a>0)(h1:b>0)(h2:c>0):
1/(a^3+b^3+a*b*c)+1/(b^3+c^3+a*b*c)+1/(c^3+a^3+a*b*c)<=1/(a*b*c)

Try these :
nth_rw 1 [\l mul_one (a*b*c)]    #Replace abc by 1*abc
ring_nf                          #Simplify by ring axiom
field_simp                       #Simplify by field axiom
refine' le_of_eq _               #Proving inequality by equality
rw [one_div]                     #Replace 1/x by x^(−1)
nth_rw 3 [\l mul_one(a*b*c)]     #Replace abc by 1*abc
rw [le_div_iff']                 #Multiply abc on both sides
────────────────────────────────────────────────────────────

IMO−1995−P2 :

theorem (a b c:R)(h0:a>0)(h1:b>0)(h2:c>0)(h3: a*b*c=1):
1/(c^3*3*(a+b))+1/(b^3*3*(a+c))+1/(a^3*3*(b+c)) >= 3/2

Try these :
refine' le_of_eq _               #Proving inequality by equality
norm_num                         #Normalize numerical expressions
rw [\l h3]                       #Replace 1 by abc
field_simp                       #Simplify by field axiom
ring_nf                          #Simplify by ring axiom
field_simp [h1, h2]              #Simplify by field axiom + h1,h2
push_cast                        #Move certain coercions inward
```

Figure 9: **Tactics suggested by LeanCopilot to two problems, namely USAMO-1997-P5 and IMO-1995-P2.**

### 2.5.4   10 Problems Solved by Our AIPS

When proving an inequality, AIPS first homogenizes both sides using the given conditions if the inequality is not already homogenized, thereby obtaining a new inequality. It then performs mixed reasoning on the new inequality to complete the proof. We present the proofs for the 10 problems solved by our AIPS as follows.

## 1. Solution to IMO-1990-Shortlist Problem

By `<function try_homo>`, It is equivalent to prove

$$\frac{a^3}{b+c+d} + \frac{b^3}{a+c+d} + \frac{c^3}{a+b+d} + \frac{d^3}{a+b+c} \geq \frac{ab}{3} + \frac{ad}{3} + \frac{bc}{3} + \frac{cd}{3}$$

by `<function check_AM_GM_Mul2>`, it remains to prove

$$\frac{a^2}{3} + \frac{b^2}{3} + \frac{c^2}{3} + \frac{d^2}{3} \leq \frac{a^3}{b+c+d} + \frac{b^3}{a+c+d} + \frac{c^3}{a+b+d} + \frac{d^3}{a+b+c}$$

by `<function try_together_l>`, it remains to prove

$$\frac{a^2+b^2+c^2+d^2}{3} \leq \frac{a^3}{b+c+d} + \frac{b^3}{a+c+d} + \frac{c^3}{a+b+d} + \frac{d^3}{a+b+c}$$

we use Hölder's inequality:

$$(a^2+b^2+c^2+d^2)^2 \leq$$
$$(a(b+c+d) + b(a+c+d) + c(a+b+d) + d(a+b+c)) \times$$
$$(a^3/(b+c+d) + b^3/(a+c+d) + c^3/(a+b+d) + d^3/(a+b+c)).$$

It remains to prove

$$\frac{a^2+b^2+c^2+d^2}{3} \leq \frac{\left(a^2+b^2+c^2+d^2\right)^2}{a\left(b+c+d\right) + b\left(a+c+d\right) + c\left(a+b+d\right) + d\left(a+b+c\right)}$$

by `<function all_cyc_mul_expr>`, it remains to prove

$$\frac{1}{3} \leq \frac{a^2+b^2+c^2+d^2}{a\left(b+c+d\right) + b\left(a+c+d\right) + c\left(a+b+d\right) + d\left(a+b+c\right)}$$

For $f(x) = x^2$, $f''(x) > 0$ for $0 < x$. we use Jensen's inequality:

$$4(a/4 + b/4 + c/4 + d/4)^2 \leq a^2 + b^2 + c^2 + d^2,$$

it remains to prove

$$\frac{1}{3} \leq \frac{4\left(\frac{a}{4} + \frac{b}{4} + \frac{c}{4} + \frac{d}{4}\right)^2}{a\left(b+c+d\right) + b\left(a+c+d\right) + c\left(a+b+d\right) + d\left(a+b+c\right)}$$

For $f(x) = x(a+b+c+d-x)$, $f''(x) < 0$ for $0 < x < a+b+c+d$, we use Jensen's inequality:

$$a(b+c+d) + b(a+c+d) + c(a+b+d) + d(a+b+c) \leq$$
$$4(a/4 + b/4 + c/4 + d/4)(3a/4 + 3b/4 + 3c/4 + 3d/4)$$

it remains to prove

$$\frac{1}{3} \leq \frac{\frac{a}{4} + \frac{b}{4} + \frac{c}{4} + \frac{d}{4}}{\frac{3a}{4} + \frac{3b}{4} + \frac{3c}{4} + \frac{3d}{4}}$$

by `<function try_simp_r>`, this is true!

**2. Solution to IMO-1993-Shortlist problem.**

To prove

$$\frac{a}{b+2c+3d} + \frac{b}{3a+c+2d} + \frac{c}{2a+3b+d} + \frac{d}{a+2b+3c} \geq \frac{2}{3}$$

we use Hölder's inequality:

$$(a+b+c+d)^2 \leq \left(\frac{a}{b+2c+3d}\right) + \frac{b}{3a+c+2d} + \frac{c}{2a+3b+d} + \frac{d}{a+2b+3c}\right) \times$$
$$(a(b+2c+3d) + b(3a+c+2d) + c(2a+3b+d) + d(a+2b+3c)).$$

It remains to prove

$$\frac{2}{3} \leq \frac{(a+b+c+d)^2}{4ab+4ac+4ad+4bc+4bd+4cd}$$

by `<function all_cyc_mul_expr>`, it remains to prove

$$\frac{2}{3(a+b+c+d)^2} \leq \frac{1}{4ab+4ac+4ad+4bc+4bd+4cd}$$

by `<function try_expand_l>`, it remains to prove

$$\frac{2}{3a^2+6ab+6ac+6ad+3b^2+6bc+6bd+3c^2+6cd+3d^2} \leq$$
$$\frac{1}{4ab+4ac+4ad+4bc+4bd+4cd}$$

by `<function nodiv_expr>`, it remains to prove

$$8ab+8ac+8ad+8bc+8bd+8cd \leq 3a^2+6ab+6ac+6ad+3b^2+6bc+6bd+3c^2+6cd+3d^2$$

by `<function zero_side>`, it remains to prove

$$0 \leq 3a^2-2ab-2ac-2ad+3b^2-2bc-2bd+3c^2-2cd+3d^2$$

by `<function check_AM_GM_Mul2>`, it remains to prove

$$0 \leq 2a^2-2ab-2ad+2b^2-2bc+2c^2-2cd+2d^2$$

by `<function check_AM_GM_Mul2>`, this is true!

---

**3. Solution to IMO-1995-P2**

By `<function try_homo>`, it is equivalent to prove

$$\frac{a^2b^2}{c(a+b)} + \frac{b^2c^2}{a(b+c)} + \frac{a^2c^2}{b(a+c)} \geq \frac{3a^{\frac{2}{3}}b^{\frac{2}{3}}c^{\frac{2}{3}}}{2}$$

We use Hölder's inequality:

$$\frac{ab+bc+ca}{2} \leq (c(a+b)+a(b+c)+b(c+a))\left(\frac{a^2b^2}{c(a+b)} + \frac{b^2c^2}{a(b+c)} + \frac{a^2c^2}{b(a+c)}\right).$$

It remains to prove

$$\frac{3a^{\frac{2}{3}}b^{\frac{2}{3}}c^{\frac{2}{3}}}{2} \leq \frac{ab+bc+ca}{2}$$

by `<function check_AM_GM>`, this is true!

---

**4. Solution to USAMO-1997-P5.**

To prove

$$\frac{1}{abc+b^3+c^3} + \frac{1}{a^3+abc+c^3} + \frac{1}{a^3+abc+b^3} \leq \frac{1}{abc}$$

by `<function check_SimpMuirhead>`, it remains to prove

$$\frac{1}{abc+b^2c+bc^2} + \frac{1}{a^2c+abc+ac^2} + \frac{1}{a^2b+ab^2+abc} \leq \frac{1}{abc}$$

by `<function try_together_l>`, this is true!

**5. Solution to 2001-IMO-P2.**
To prove
$$\frac{a}{\sqrt{a^2 + 8bc}} + \frac{b}{\sqrt{8ac + b^2}} + \frac{c}{\sqrt{8ab + c^2}} \geq 1,$$
we use Hölder's inequality:
$$(a + b + c)^3 \leq$$
$$\left(\frac{a}{\sqrt{a^2 + 8bc}} + \frac{b}{\sqrt{8ac + b^2}} + \frac{c}{\sqrt{8ab + c^2}}\right)^2 \left(a(a^2 + 8bc) + b(8ac + b^2) + c(8ab + c^2)\right).$$
It remains to prove
$$1 \leq \frac{(a + b + c)^{\frac{3}{2}}}{\sqrt{a^3 + 24abc + b^3 + c^3}}.$$
by `<function all_cyc_mul_expr>`, it remains to prove
$$\sqrt{a^3 + 24abc + b^3 + c^3} \leq (a + b + c)^{\frac{3}{2}}$$
by `<function no_pow>`, it remains to prove
$$a^3 + 24abc + b^3 + c^3 \leq (a + b + c)^3$$
by `<function zero_side>`, it remains to prove
$$0 \leq -a^3 - 24abc - b^3 - c^3 + (a + b + c)^3$$
by `<function try_expand_r>`, it remains to prove
$$0 \leq 3a^2 b + 3a^2 c + 3ab^2 - 18abc + 3ac^2 + 3b^2 c + 3bc^2$$
by `<function check_AM_GM>`, it remains to prove
$$0 \leq 3a^2 b - 9abc + 3ac^2 + 3b^2 c$$
by `<function check_AM_GM>`, this is true!

---

**6. Solution to USAMO-2003-P5.**
To prove
$$\frac{(a + b + 2c)^2}{2c^2 + (a + b)^2} + \frac{(a + 2b + c)^2}{2b^2 + (a + c)^2} + \frac{(2a + b + c)^2}{2a^2 + (b + c)^2} \leq 8$$
we have
$$f(x) = \frac{(x + 1)^2}{(1 - x)^2 + 2x^2} \leq \frac{12x + 4}{3} \text{ for } 0 < x < 1$$
$$\iff -\frac{(3x - 1)^2 \cdot (4x + 1)}{3 \cdot (3x^2 - 2x + 1)} \leq 0 \text{ for } 0 < x < 1,$$
which is true.
Substitute $x$ for $\dfrac{c}{a + b + c}$, we have
$$\frac{(a + b + 2c)^2}{2c^2 + (a + b)^2} \leq \frac{4c}{a + b + c} + \frac{4}{3}$$
It remains to prove
$$\frac{4a}{a + b + c} + \frac{4b}{a + b + c} + \frac{4c}{a + b + c} + 4 \leq 8$$
by `<function try_together_1>`, this is true!

**7. Solution to Polish-2004 Problem**

We use Hölder's inequality:

$$(a+b+c+d)^4 \leq \left(\frac{a}{(a^3+63bcd)^{\frac{1}{3}}} + \frac{b}{(63acd+b^3)^{\frac{1}{3}}} + \frac{c}{(63abd+c^3)^{\frac{1}{3}}} + \frac{d}{(63abc+d^3)^{\frac{1}{3}}}\right)^3 \times$$

$$(a(a^3+63bcd) + b(b^3+63acd) + c(c^3+63abd) + d(d^3+63abc)).$$

It remains to prove

$$1 \leq \frac{(a+b+c+d)^{\frac{4}{3}}}{(a^4+252abcd+b^4+c^4+d^4)^{\frac{1}{3}}},$$

by `<function no_pow>`, it remains to prove

$$1 \leq \frac{(a+b+c+d)^4}{a^4+252abcd+b^4+c^4+d^4},$$

by `<function nodiv_expr>`, it remains to prove

$$a^4 + 252abcd + b^4 + c^4 + d^4 \leq (a+b+c+d)^4,$$

by `<function zero_side>`, it remains to prove

$$0 \leq -a^4 - 252abcd - b^4 - c^4 - d^4 + (a+b+c+d)^4$$

by `<function try_expand_r>`, it remains to prove

$$0 \leq 4a^3b + 4a^3c + 4a^3d + 6a^2b^2 + 12a^2bc + 12a^2bd + 6a^2c^2 + 12a^2cd + 6a^2d^2 + 4ab^3 \ldots$$

by `<function check_AM_GM>`, it remains to prove

$$0 \leq 4a^3b + 4a^3c + 4a^3d + 6a^2b^2 + 12a^2bc + 12a^2bd + 12a^2cd + 6a^2d^2 + 4ab^3 \ldots$$

by `<function sep_neg>`, it remains to prove

$$216abcd \leq 4a^3b + 4a^3c + 4a^3d + 6a^2b^2 + 12a^2bc + 12a^2bd + 12a^2cd + 6a^2d^2 \ldots$$

by `<function check_AM_GM>`, it remains to prove

$$216abcd \leq 4a^3b + 4a^3c + 4a^3d + 12a^2bc + 12a^2bd + 12a^2cd + 4ab^3 + \ldots$$

by `<function check_AM_GM>`, it remains to prove

$$216abcd \leq 4a^3b + 4a^3c + 12a^2bc + 12a^2bd + 12a^2cd + 12ab^2c + 12ab^2d + 12abc^2 + \ldots$$

by `<function check_AM_GM>`, it remains to prove

$$216abcd \leq 4a^3b + 4a^3c + 12a^2bc + 12a^2cd + 12ab^2d + 12abc^2 + 88abcd + 12abd^2 + \ldots$$

by `<function check_AM_GM>`, it remains to prove

$$216abcd \leq 4a^3b + 4a^3c + 12a^2bc + 136abcd + 12abd^2 + 4ac^3 + 12ac^2d$$
$$+4ad^3 + 4b^3c + 4b^3d + 12b^2cd + 4bd^3 + 4c^3d$$

by `<function check_AM_GM>`, it remains to prove

$$216abcd \leq 4a^3b + 4a^3c + 184abcd + 4ac^3 + 4ad^3 + 4b^3c + 4b^3d + 4bd^3 + 4c^3d$$

by `<function zero_side>`, it remains to prove

$$0 \leq 4a^3b + 4a^3c - 32abcd + 4ac^3 + 4ad^3 + 4b^3c + 4b^3d + 4bd^3 + 4c^3d$$

by `<function check_AM_GM>`, it remains to prove

$$0 \leq 4a^3b - 16abcd + 4ad^3 + 4b^3c + 4c^3d$$

by `<function check_AM_GM>`, this is true!

**8. Solution to USA-IMO-Team-Selection-2010-P2.**
By `<function try_homo>`, it is equivalent to prove

$$\frac{a^3 b^3}{c^2 (a + 2b)^2} + \frac{a^3 c^3}{b^2 (2a + c)^2} + \frac{b^3 c^3}{a^2 (b + 2c)^2} \geq \frac{a^{\frac{2}{3}} b^{\frac{2}{3}} c^{\frac{2}{3}}}{3}$$

we use Hölder's inequality:

$$(ab + ac + bc)^3 \leq (a(b + 2c) + b(2a + c) + c(a + 2b))^2 \times$$

$$(a^3 b^3 / (c^2 (a + 2b)^2) + a^3 c^3 / (b^2 (2a + c)^2) + b^3 c^3 / (a^2 (b + 2c)^2)).$$

It remains to prove

$$\frac{a^{\frac{2}{3}} b^{\frac{2}{3}} c^{\frac{2}{3}}}{3} \leq \frac{ab}{9} + \frac{ac}{9} + \frac{bc}{9}$$

by `<function check_AM_GM>`, this is true!

---

**9. Solution to Korea-2011-P4.**
To prove

$$\frac{1}{a^2 - 4a + 9} + \frac{1}{b^2 - 4b + 9} + \frac{1}{c^2 - 4c + 9} \leq \frac{7}{18},$$

we have

$$f(x) = 1/(x^2 - 4x + 9) \leq \frac{2 + x}{18} \text{ for } 0 < x < 1$$

$$\iff \quad -\frac{x (x - 1)^2}{18 (x^2 - 4x + 9)} \leq 0 \text{ for } 0 < x < 1,$$

which is true. Substitute $x$ for $a/(a + b + c)$, we have

$$1/(a^2 - 4a + 9) = \frac{(a + b + c)^2}{a^2 - 4a (a + b + c) + 9 (a + b + c)^2} \leq \frac{3a + 2b + 2c}{18a + 18b + 18c}.$$

It remains to prove

$$\frac{3a + 2b + 2c}{18a + 18b + 18c} + \frac{2a + 3b + 2c}{18a + 18b + 18c} + \frac{2a + 2b + 3c}{18a + 18b + 18c} \leq \frac{7}{18},$$

by `<function try_together_1>`, this is true.

**10. Solution to Japan-2014-P5**

By `<function try_homo>`, it is equivalent to prove

$$\frac{a\,(a+b+c)}{9bc+4\,(b-c)^2+(a+b+c)^2}+\frac{b\,(a+b+c)}{9ac+4\,(-a+c)^2+(a+b+c)^2}+$$

$$\frac{c\,(a+b+c)}{9ab+4\,(a-b)^2+(a+b+c)^2}\geq\frac{1}{2}.$$

We use Hölder's inequality:

$$(a+b+c)^3\leq$$

$$(\frac{a\,(a+b+c)}{9bc+4\,(b-c)^2+(a+b+c)^2}+\frac{b\,(a+b+c)}{9ac+4\,(-a+c)^2+(a+b+c)^2}+\frac{c\,(a+b+c)}{9ab+4\,(a-b)^2+(a+b+c)^2})\times$$

$$\left\{a\left(9a^2+4\,(b-c)^2+(a+b+c)^2\right)+b\left(9b^2+4\,(-a+c)^2+(a+b+c)^2\right)+\right.$$

$$\left.c\left(9c^2+4\,(a-b)^2+(a+b+c)^2\right)\right\}$$

It remains to prove

$$\frac{1}{2}\leq\frac{(a+b+c)^3}{27abc+4a\,(b-c)^2+a\,(a+b+c)^2+4b\,(a-c)^2+b\,(a+b+c)^2+4c\,(a-b)^2+c\,(a+b+c)^2}$$

by `<function nodiv_expr>`, it remains to prove

$$27abc+4a\,(b-c)^2+a\,(a+b+c)^2+4b\,(a-c)^2+b\,(a+b+c)^2+4c\,(a-b)^2+c\,(a+b+c)^2$$

$$\leq 2\,(a+b+c)^3$$

by `<function zero_side>`, it remains to prove

$$0\leq-27abc-4a\,(b-c)^2-a\,(a+b+c)^2-4b\,(a-c)^2-b\,(a+b+c)^2-4c\,(a-b)^2$$

$$-c\,(a+b+c)^2+2\,(a+b+c)^3$$

by `<function try_expand_r>`, it remains to prove

$$0\leq a^3-a^2b-a^2c-ab^2+3abc-ac^2+b^3-b^2c-bc^2+c^3$$

by `<function check_schur>`, this is true!

# 3 Human Evaluation of Generated Synthetic Theorems

We select 10 synthetic problems generated by our AIPS for evaluation, and 4 IMO problems for comparison. Then, we invite three professional contestants to evaluate the difficulty and elegance of these 14 problems. Two of the evaluators are National Mathematical Olympiad gold medalists, and one is a silver medalist. The difficulty and elegance are needed to assign a score from 1 to 7, respectively.

## 3.1 10 Synthetic Theorems and 4 Comparison IMO Problems

### 3.1.1 10 Synthetic Theorems

- **(Problem1)**
  Given $a, b, c > 0$, then
  $$\frac{(a+b+c)^3}{(ab+bc+ca)^2} \leq \frac{4a}{(b+c)^2} + \frac{4b}{(c+a)^2} + \frac{4c}{(a+b)^2}$$

- **(Problem2)**
  Given $a, b, c > 0$, then
  $$\frac{27(a^2+b^2)^2(b^2+c^2)^2(c^2+a^2)^2}{(a^4+b^4+c^4+3a^2b^2+3b^2c^2+3c^2a^2)^3} \leq 1$$

- **(Problem3)**
  Given $a, b, c > 0$, then
  $$\frac{abc(a+b+c)^3}{3(ab+bc+ca)(a^3c+ab^3+bc^3)} \leq 1$$

- **(Problem4)**
  Given $a, b, c > 0$, then
  $$\frac{2a}{\sqrt{2a^2+b^2+c^2}} + \frac{2b}{\sqrt{2b^2+c^2+a^2}} + \frac{2c}{\sqrt{2c^2+a^2+b^2}} \leq \frac{3\sqrt{2}(a+b+c)}{\sqrt{5a^2+5b^2+5c^2+ab+bc+ca}}$$

- **(Problem5)**
  Given $a, b, c > 0$, then
  $$\frac{\sqrt{6}(a+b+c)^2}{6\sqrt{a^4+b^4+c^4+a^2b^2+b^2c^2+c^2a^2}} \leq \frac{a}{\sqrt{2a^2+b^2+c^2}} + \frac{b}{\sqrt{2b^2+c^2+a^2}} + \frac{c}{\sqrt{2c^2+a^2+b^2}}$$

- **(Problem6)**
  Given $a, b, c > 0$, then
  $$2(a+b+c)^{\frac{3}{2}} \leq (\sqrt{a+b}+\sqrt{b+c}+\sqrt{c+a})\sqrt{a^2+b^2+c^2+ab+bc+ca}$$

- **(Problem7)**
  Given $a, b, c > 0$, then
  $$\frac{(a^4+b^4+c^4)^{\frac{3}{2}}}{\sqrt{ab^2+bc^2+ca^2-abc}\sqrt{a+b+c}} \leq \frac{a^5}{\sqrt{ca+b^2}} + \frac{b^5}{\sqrt{ab+c^2}} + \frac{c^5}{\sqrt{bc+a^2}}$$

- **(Problem8)**
  Given $a, b, c > 0$, then
  $$\frac{54abc+(a+b+c)^3}{\left(\sqrt{a^2+2bc}+\sqrt{2ab+c^2}+\sqrt{2ac+b^2}\right)^2} \leq a+b+c$$

- **(Problem9)**
  Given $a, b, c > 0$, then
  $$\frac{a^2b}{(a+b)^3} + \frac{ac^2}{(a+c)^3} + \frac{b^2c}{(b+c)^3} \leq \frac{3}{8}$$

- **(Problem10)**
  Given $a, b, c > 0$, then

$$\frac{(ab + ac + bc)^2}{\sqrt{a^2 + b^2 + c^2}\sqrt{a^2 + b^2 + c^2 + 3ab + 3bc + 3ca}} \leq \frac{a^2b}{\sqrt{b^2 + 3ac}} + \frac{b^2c}{\sqrt{c^2 + 3ab}} +$$
$$\frac{c^2a}{\sqrt{a^2 + 3bc}}$$

### 3.1.2   4 IMO Problems

- **(1995-imo-2)**

  Given $a, b, c > 0$ and $abc = 1$, then $\dfrac{1}{c^3(a+b)} + \dfrac{1}{b^3(a+c)} + \dfrac{1}{a^3(b+c)} \geq \dfrac{3}{2}$

- **(2001-imo-2)**

  Given $a, b, c > 0$, then $\dfrac{a}{\sqrt{a^2 + 8bc}} + \dfrac{b}{\sqrt{8ac + b^2}} + \dfrac{c}{\sqrt{8ab + c^2}} \geq 1$

- **(2006-imo-3)**

  Assume $a, b, c$ are three real numbers, then $|ab(a^2 - b^2) + bc(b^2 - c^2) + ca(c^2 - a^2)| \leq \dfrac{9}{16\sqrt{2}}(a^2 + b^2 + c^2)^2$

- **(2020-imo-2)**

  Assume $a \geq b \geq c \geq d \geq 0$ and $a + b + c + d = 1$, prove that $a^a b^b c^c d^d (a + 2b + 3c + 4d) < 1$

### 3.2   Human Evaluation Results

The rating scores by the three professional contestants are reported in Table 1. The third expert does not assign scores to the four IMO problems, believing the average difficulty of the ten problems is significantly lower than that of IMO problems. The first expert does not give a difficulty score for Problem 8 because he does not solve it. From the table, we observe that while the average difficulty does not compare with IMO inequalities, a few problems, such as Problem 9 and Problem 7, reach the IMO level.

Table 1: Scores given by human experts on synthetic theorems and IMO problems. Scores range from 1 to 7. **GM** denotes gold medalist, and **SM** denotes silver medalist.

| Problem | Expert 1 (GM) | | Expert 2 (GM) | | Expert 3 (SM) | |
|---|---|---|---|---|---|---|
| | **Difficulty** | **Elegance** | **Difficulty** | **Elegance** | **Difficulty** | **Elegance** |
| 1 | 2 | 2 | 2 | 3 | 1 | 2.5 |
| 2 | 1 | 1 | 1 | 2 | 1 | 1 |
| 3 | 2 | 1 | 4 | 2 | 1.5 | 1 |
| 4 | 3 | 2 | 3 | 2 | 2 | 1.5 |
| 5 | 2 | 1 | 2 | 2 | 1.5 | 1 |
| 6 | 2 | 2 | 2 | 2 | 1.5 | 1.5 |
| 7 | 5 | 1 | 4 | 2 | 2 | 2 |
| 8 | NA | 2 | 3 | 2 | 1 | 1.5 |
| 9 | 4 | 3 | 4 | 5 | 2.5 | 2 |
| 10 | 4 | 1 | 3 | 1 | 1 | 1.5 |
| IMO-1995-2 | 2 | 4 | 3 | 5 | NA | NA |
| IMO-2001-2 | 3 | 4 | 3 | 5 | NA | NA |
| IMO-2006-3 | 3 | 3 | 5 | 3 | NA | NA |
| IMO-2020-2 | 2 | 2 | 4 | 3 | NA | NA |

### 3.3 Synthetic Theorem Selected for Mathematical Olympiad

Among the 10 synthetic problems above, problem 4 was chosen as a competition problem in a major city's 2024 Mathematical Olympiad, as shown in Fig. 10. It received positive feedback for its appropriate difficulty, concise form, and variety of solutions. This problem was posted online, and 75 contestants provided their evaluations on its difficulty and elegance. The score distributions are shown in Fig. 11. The average difficulty score was 3.3 out of 7, and the elegance score was 2.2 out of 5. The 4 solutions to this problem, including one provided by our AIPS and 3 solutions collected from the competition organizers, are given as follows.

> **Problem:** Given three positive real numbers $a, b, c$, prove that
> $$\frac{2a}{\sqrt{2a^2 + b^2 + c^2}} + \frac{2b}{\sqrt{2b^2 + c^2 + a^2}} + \frac{2c}{\sqrt{2c^2 + a^2 + b^2}} \leq \frac{3\sqrt{2}(a + b + c)}{\sqrt{5a^2 + 5b^2 + 5c^2 + ab + bc + ca}}$$

Figure 10: Selected theorem for a major city's Mathematical Olympiad.

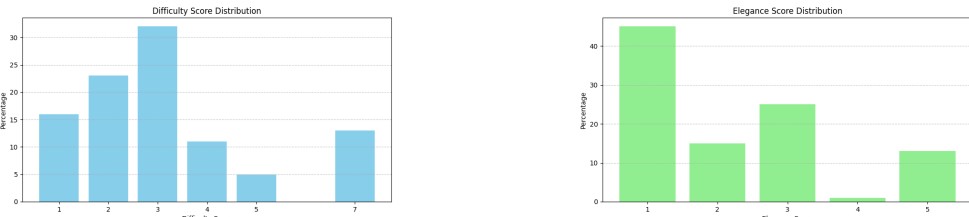

Figure 11: Score distributions evaluated by 75 contestants online.

**Proof 1. (Modified from AIPS' proof)**

$f''(x) = \dfrac{6x(-a^2 - b^2 - c^2)}{(a^2 + b^2 + c^2 + x^2)^{\frac{5}{2}}} < 0$ for $x$ satisfying $0 < x < a^2 + b^2 + c^2$, where $f(x) =$

$\dfrac{2x}{\sqrt{x^2 + a^2 + b^2 + c^2}}$. By Jensen's inequality, **LHS** $\leq 3 \cdot \dfrac{2 \cdot \frac{a+b+c}{3}}{\sqrt{a^2 + b^2 + c^2 + \left(\frac{a+b+c}{3}\right)^2}}$. It

suffices to prove

$$3 \cdot \frac{2 \cdot \frac{a+b+c}{3}}{\sqrt{a^2 + b^2 + c^2 + \left(\frac{a+b+c}{3}\right)^2}} \leq \frac{3\sqrt{2}(a + b + c)}{\sqrt{5a^2 + 5b^2 + 5c^2 + ab + bc + ca}}.$$

Expanding the left-hand side, this is true. $\qquad\square$

---

**Proof 2. (Given by Humans)**

Without loss of generality, assume $a \geq b \geq c$ and $a^2 + b^2 + c^2 = 1$. Then the inequality in question is equivalent to

$$\sum \frac{a}{\sqrt{1 + a^2}} \leq \frac{3(a + b + c)}{\sqrt{9 + (a + b + c)^2}}$$

Notice that

$$\frac{a}{\sqrt{1 + a^2}} = \sqrt{1 - \frac{1}{1 + a^2}} \geq \sqrt{1 - \frac{1}{1 + b^2}} = \frac{b}{\sqrt{1 + b^2}}$$

By Chebyshev inequality, we get

$$\left(\sum \sqrt{1 + a^2}\right)\left(\sum \frac{a}{\sqrt{1 + a^2}}\right) \leq 3(a + b + c).$$

Then it suffices to prove

$$\sum \sqrt{1 + a^2} \geq \sqrt{9 + (a + b + c)^2}$$

which is equivalent to show $6 + 2\sum ab \leq 2\sum \sqrt{1 + a^2}\sqrt{1 + b^2}$. Notice that

$$1 + ab \leq \sqrt{1 + a^2}\sqrt{1 + b^2} \iff 2ab \leq a^2 + b^2$$

and the right-hand-side holds by AM-GM inequality. Therefore we have finished the proof. $\qquad\square$

**Proof 3. (Given by Humans)**

First we divide the proof into two subgoals:

$$\frac{3\sqrt{2}(a+b+c)}{\sqrt{5a^2+5b^2+5c^2+ab+bc+ca}} \geq \frac{2(a+b+c)}{\sqrt{\frac{4}{3}(a^2+b^2+c^2)}} \tag{1}$$

and

$$\sum \frac{2a}{\sqrt{\frac{4}{3}(a^2+b^2+c^2)}} \geq \sum \frac{2a}{\sqrt{2a^2+b^2+c^2}} \tag{2}$$

Where $\sum$ denotes cyclic summation. The proof of (1) follows from the fact that $a^2+b^2+c^2 \geq ab+bc+ca$. For the second part, we apply Chebyshev's inequality.

Without loss of generality, we assume $a \geq b \geq c$. First notice that

$$\sum \frac{2a}{\sqrt{\frac{4}{3}(a^2+b^2+c^2)}} - \frac{2a}{\sqrt{2a^2+b^2+c^2}} = \frac{1}{3}\sum x_a(2a^2-b^2-c^2) \tag{3}$$

where

$$x_a = \frac{2a}{\sqrt{\frac{4}{3}(a^2+b^2+c^2)}\sqrt{2a^2+b^2+c^2}(\sqrt{\frac{4}{3}(a^2+b^2+c^2)}+\sqrt{2a^2+b^2+c^2})}$$

and $x_b$, $x_c$ are defined similarly. We claim that $x_a \geq x_b \geq x_c$. For $x_a \geq x_b$, it suffice to show two inequalities:

$$a\sqrt{a^2+2b^2+c^2} \geq b\sqrt{2a^2+b^2+c^2}$$

$$a(a^2+2b^2+c^2) \geq b(2a^2+b^2+c^2)$$

Both can be proven by factorization, and the proof of $x_b \geq x_c$ is similar.

Since $a \geq b \geq c$, we get $2a^2-b^2-c^2 \geq 2b^2-c^2-a^2 \geq 2c^2-a^2-b^2$. Combining with $x_a \geq x_b \geq x_c$ and applying Chebyshev's inequality, we get $\sum x_a(2a^2-b^2-c^2) \geq 0$.

Finally, combining with (3), we conclude that (2) is proved. $\square$

---

**Proof 4. (Given by Humans)**

Let $S = a^2+b^2+c^2$ and $t = \dfrac{a+b+c}{3}$. Substituting into the inequality and rearranging:

$$\text{LHS} = \sum \frac{2a}{\sqrt{S+a^2}}$$

$$\text{RHS} = \sum\left((2(t^2+S)^{-\frac{1}{2}} - 2t^2(t^2+S)^{-\frac{3}{2}})(a-t) + 2t(t^2+S)^{-\frac{1}{2}}\right)$$

It suffice to show

$$\frac{2a}{\sqrt{a^2+S}} \leq \frac{2S(a-t)+2t(t^2+S)}{(t^2+S)^{\frac{3}{2}}}$$

which is equivalent to

$$3a^2t^4S + 3a^2t^2S^2 \leq (2Sa^3t^3 + St^6) + (2S^2ta^3 + S^2a^4)$$

The last inequality is proved by applying AM-GM inequality. $\square$