# OpenReview forum: "Proving Olympiad Algebraic Inequalities without Human Demonstrations"
_NeurIPS.cc/2024/Datasets_and_Benchmarks_Track — NeurIPS 2024 Track Datasets and Benchmarks Poster_

### Official Review · Reviewer_V585 · 2024-07-17

**Rating:** 8
**Confidence:** 4
**Correctness:** Yes.
**Clarity:** Yes.

**Review:**

Overall, the submission presents an impressive work and a major advance in AI-aided theorem proving. It is amazing that AIPS is capable of solving several olympiad-level inequalities via a systematic integration of several key AI techniques. However, the data is not available at the current point. Assuming the data will be made available upon acceptance, I give the current rating of clear acceptance.

**Strengths:**

1. A novel and strong neuro-symbolic proof system for the challenging inequality-proving scenario. Compared to existing symbolic or LLM-based systems, the proposed system smartly combines the advantages from both worlds and demonstrates strong empirical performance. It advances the frontiers of AI systems.

2. The proposed proof system is elegant and represents the principled way of leveraging existing AI methods spanning symbolic computing, reasoning, and neuro methods. Moreover, the synthetic dataset would be of great benefit to the community (as long as it is made public).

3. The writing quality is high and most parts are very clear. It is a pleasure read.

**Additional Feedback:**

N/A

**Documentation:**

Partly. Though the appendix provides many details and examples, the code, full synthetic dataset, and the benchmark set are not available with clear documentation.

**Ethics:**

No.

**Limitations:**

The data and code are not open source. The provided URL links to a blank webpage.

-------

The author's rebuttal now provides a correct link, and the concern is resolved.

**Opportunities For Improvement:**

1. It would be great to mention more details of the value network finetuning. I found them in the appendix, but they are very brief in the main text. Especially, the value iteration in the fine-tuning process is ambiguous (line 220-229). Why the values on the proof path are scaled down? Does a lower value represent a higher chance to be selected or the opposite?

2. The evaluation benchmark is a bit limited since it only contains 20 questions. It is somewhat reasonable since high-quality and challenging Olympic-level inequalities are hard to find. But some exercise books or more national olympiad games could be sources to expand the benchmark set.


Minor:
1. Line 189: What is the "homogenization" procedure mentioned here?

**Relation To Prior Work:**

Yes.

**Summary And Contributions:**

The submission introduces a novel inequality-proving system AIPS based on proof search and value network learning. The system integrates a symbolic deductive engine and pattern matching for synthetic theorem generation. Based on the 191643 synthesized inequality, the system combines deductive engine and a value network for efficient candidate selection and proof exploration. Evaluation of 20 Olympiad-level inequalities (where it solves 10) demonstrates the effectiveness.

---

> ### Author Rebuttal · Authors · 2024-08-15
>
> Dear reviewer,
>
> Thank you for your positive evaluation and constructive feedback! Below we address your questions and concerns. Please feel free to follow up if you have further questions.
>
> ## More details of the value network finetuning. Why are the values on the proof path scaled down? Does a lower value represent a higher chance to be selected or the opposite?
>
> The procedure of value network finetuning is described as follows. After AIPS successfully proves a theorem, each node along the proof path is relabeled with a value that is smaller than the original value, while the value of nodes that have been searched but not part of the proof path are increased. These updated labels are then used to fine-tune the value network before moving on to the next problem. After fine-tuning, AIPS is able to solve 10 out of 20 problems in the test set and demonstrates increased efficiency by finding proofs with fewer search loops.
>
> We can see that, according to the node value updating strategy defined in value network finetuning, nodes with lower values are more likely to be selected in the next search loop. The primary purpose of designing the value network is to assess the difficulty of each inequality (with lower value indicating less difficulty), helping AIPS to prioritize the subgoal that is most likely to be solved for the next search iteration.
>
> ## Small size of the test set and test-set expansion.
> We acknowledge that the current test set, consisting of 20 IMO-level problems, is somewhat limited in size. This is primarily due to the scarcity of challenging IMO-level inequalities, which constitute about 30% of algebra problems in IMO-level contests, while algebra problems themselves account for approximately 25% of the total problems in mathematical competitions.
>
> Thanks to your valuable suggestions, we will expand our evaluation benchmark by collecting more problems from exercise books and additional national Olympiad competitions, though the average difficulty of these problems may not reach the IMO level.
>
> Additionally, our model can automatically generate IMO-level problems, as detailed in Section 3 of our supplementary material. Therefore we will also generate a large test dataset with varying difficulty levels, some of which will meet IMO-level standards. This new synthetic test set will be made publicly available on the official AIPS website (https://sites.google.com/view/aips2).
>
> ## Homogenization Procedure in Theorem Proving
> Homogenization is a concept in mathematics. The homogenization procedure in our paper means homogenizing an algebraic expression or inequality by incorporating the given conditions in the problem. For example, consider this problem: given $a,b,c>0$ and $abc=1$, prove that $a+b+c\geq3$. AIPS detects that the degree of the left-hand side (LHS) of the conclusion $a+b+c\geq3$ is $1$, while the right-hand side (RHS) has a degree of $0$. By using the condition $abc=1$, we can express $3$ as $3(abc)^{\frac{1}{3}}$ and substitute this into the RHS. This homogenizes the inequality, allowing AIPS to prove the theorem without requiring to consider the condition $abc=1$.
>
> ## Correction and access to AIPS data resources.
> We apologize for the URL typos provided in the submitted version. The correct link to AIPS data resources is https://sites.google.com/view/aips2.
>
> A large synthetic theorem dataset automatically generated by AIPS is provided on this website. Additionally, this dataset has been converted into Lean format by another researcher, which can be obtained from the Hugging Face platform.

---

> > ### Comment · Reviewer_V585 · 2024-08-29
> >
> > Thanks for the response. My concerns are addressed and I will keep my current score.

---

### Official Review · Reviewer_5Jfo · 2024-07-23
**May not very suitable for dataset and benchmark track.**

**Rating:** 3
**Confidence:** 3
**Correctness:** See above.
**Clarity:** See above.

**Review:**

The reviewer appreciate the contributions by the AIPS system. However, the reviewer cannot align the major contributions (which seems to be methodological as described in the introduction part by the authors) with the CFP of the dataset and benchmark track:

- **New datasets**, or carefully and thoughtfully designed (collections of) datasets based on previously available data.
- **Data generators** and reinforcement learning environments.
- Data-centric AI methods and tools, e.g. to **measure and improve data quality or utility, or studies in data centric AI that bring important new insight**.
- Advanced **practices in data collection and curation** that are of general interest even if the data itself cannot be shared.
- **Frameworks for responsible dataset development**, audits of existing datasets, identifying significant problems with existing datasets and their use
- **Benchmarks on new or existing datasets, as well as benchmarking tools**.
- In-depth **analyses of machine learning challenges and competitions** (by organisers and/or participants) that yield important new insight.
- **Systematic analyses of existing systems** on novel datasets yielding important new insight.


Due to the mismatch of the theme, the reviewer would like to suggest a desk rejection.

**Strengths:**

See above.

**Additional Feedback:**

See above.

**Documentation:**

See above.

**Ethics:**

See above.

**Limitations:**

See above.

**Opportunities For Improvement:**

See above.

**Relation To Prior Work:**

See above.

**Summary And Contributions:**

This paper proposes an Algebraic Inequality Proving System (AIPS) to solve the algebraic inequalities. The contributions are 1. the AIPS engine focuses on ternary and quaternary inequalities; 2. the value network trained by curriculum learning benefits symbolic algebraic inequality proving; 3. The AIPS reaches the state-of-the-art performance in 20 IMO-level inequalities.

---

> ### Author Rebuttal · Authors · 2024-08-15
>
> Below we list the detailed facts that illustrate how our work aligns well with the scope of the dataset and benchmark track. We hope this helps address any potential misunderstandings and encourages the reviewer to reassess the contributions our work brings to the field.
>
> ## (1) Alignment with the Call for Papers on "New datasets"
>
> Our work introduces a **new**, **high-quality**, **large-scale dataset** of algebraic inequalities, directly aligning with the first point of the CFP. As mentioned in lines 162-163, Section 3.2, our AIPS deductive engine has generated a theorem dataset containing over 190,000 inequalities, surpassing the number of theorems found in formal libraries such as Mathlib4 [1] and CoqGym [2]. Some theorems generated by AIPS, which have undergone careful evaluation by professional contestants, are considered to reach the International Mathematical Olympiad (IMO) level (Please refer to Sections 3.1 and 3.2 in the supplementary material). Notably, one theorem was selected as a competition problem for a major city’s 2024 Mathematical Olympiad (Please refer to Section 3.3 in the supplementary material).
>
> The generated theorem dataset is publicly released on this website: https://sites.google.com/view/aips2.
>
> Additionally, this dataset has been converted into Lean format by another researcher, which can be obtained from the Hugging Face platform.
>
> Due to the lack of large-scale, high-quality datasets, solving Olympiad-level mathematical problems poses great challenges to existing AI systems. Our proposed theorem dataset will significantly alleviate the data bottleneck in the field of AI4Math and push forward the complex reasoning capability of intelligent systems.
>
> ## (2) Synthetic Theorem Data Generator
>
> We propose a synthetic theorem data generator based on our symbolic deductive engine for algebra, which aligns with the second point of the CFP on "**Data generators**".
>
> We have summarized this contribution point in lines 70-72, stating that “We propose a symbolic deductive engine capable of efficiently generating high-quality and solving high-difficulty algebraic inequality theorems.” The implementation procedure of this theorem data generator is described in Algorithm 1 (see the details on Page 5). Using this generator, we have generated a synthetic dataset of over 190,000 theorems on 32 CPUs for 8 hours. During the period of rebuttal, we generated another large test dataset with varying difficulty levels to expand the test set. Soon, we will release the code of this data generator on the project website.
>
> ## (3) A new benchmark
>
> We have proposed an Algebraic Inequality Proving System (AIPS) and evaluated the main theorem proving methods on a test set of 20 IMO-level inequality problems, establishing a new benchmark for the field. This aligns with the sixth point of the CFP “**Benchmarks on new or existing datasets**.”
>
> AIPS provides new insights into how algebraic inequality datasets can be leveraged to train a pioneering AI system capable of solving IMO-level inequalities, marking a significant breakthrough in the field of AI4Math, as well as providing a benchmark for future theorem provers.
>
> ## (4) Summary:
>    - First, AIPS is, to our knowledge, the first AI system capable of generating Olympiad-level algebraic inequality theorems, with some reaching IMO-level difficulty. This generation process is highly efficient and helps alleviate the data bottleneck in the AI4Math field.
>    - Second, our work introduces a large, high-quality theorem dataset and a synthetic theorem data generator, both of which will contribute to advancing the AI4Math field by enabling the training of models to tackle complex math problems.
>    - Finally, our proving system successfully proves 10 out of 20 IMO-level inequalities, outperforming state-of-the-art methods. This achievement not only establishes a benchmark for algebraic inequality proving but also serves as a model for addressing similarly challenging mathematical problems.
>
> [1] https://github.com/leanprover-community/mathlib4
> [2] https://github.com/princeton-vl/CoqGym

---

### Official Review · Reviewer_gDey · 2024-07-24
**Proving Olympiad Algebraic Inequalities without Human Demonstrations**

**Rating:** 7
**Confidence:** 3
**Correctness:** Yes.
**Clarity:** yes.

**Review:**

Pros:
1. This is a complex and understudied problem, and this paper represents a step towards evaluating the reasoning and analytical abilities of models over algebraic inequalities.
2. The integration of symbolic solving and neural reasoning is done in a well defined and properly justified manner.
3. The evaluation is well designed, with insights into some of the failures of some models.
4. The AIPS benchmark allows for the synthesis of theorems that allows for generating training and finetuning data without needing to actually source the data from actual problems.

Questions and Suggestions:
1. The size of the test set is quite small, though that is understandable since obtaining this set is costly in terms of time and manual effort. However, a discussion into extending this test set would serve well.
2. What are the architectures of the value neural network in the neural prover?
3. What is the difference between 'AIPS with pretrained value network' and 'AIPS'? What data is the network pretrained on and how does it differ from the plain 'AIPS'?
4. How does pattern matching of a theorem occur with an expression tree such as shown in Figure 1? Specifically, how are the theorems such as the AM-GM inequality represented within the system in order to conduct the pattern matching?
5. In algorithm 1, what are sets 'S' and 'O', and where are they obtained from?
6. Are there any limitations? It would be useful to have a limitations section seeing that there is still some space left.

**Strengths:**

See pros.

**Additional Feedback:**

NA.

**Documentation:**

Yes.

**Ethics:**

No.

**Limitations:**

It would be nice to have a limitations section that addresses the issues raised in the review if not already addressed within the rest of the paper.

**Opportunities For Improvement:**

See questions and suggestions.

**Relation To Prior Work:**

Yes.

**Summary And Contributions:**

This paper proposes the AIPS system to prove complex Olympiad-level algebraic inequalities which involve several complex steps of reasoning. The paper proposes a system consisting of a symbolic deductive engine, synthetic theorem generation, and the neural prover.
It also evaluates the AIPS system with 10 models, including LLMs and Neurosymbolic techniques over 20 IMO-level problems.

---

> ### Author Rebuttal · Authors · 2024-08-15
>
> Dear Reviewer,
>
> Thank you for taking the time to review our paper! Your feedback is truly valuable, and we address your questions and concerns below. Please feel free to follow up if you have further questions.
>
> ## The Small Size of the Test Set and Consideration About its Expansion
> We acknowledge that the current test set (20 IMO-level problems) is relatively small. This arises from the difficulty in collecting challenging inequality problems. They constitute about 30% of algebra problems in IMO-level contests, while algebra problems themselves account for about 25% of the total problems in Mathematical Olympiad, which also include geometry, number theory, and combinatorics problems.
>
> Thanks to the reviewers' valuable suggestions, we will expand our test set by collecting more problems from exercise books and national Olympiads, though their average difficulty may not reach IMO level.
>
> Additionally, our model can generate IMO-level problems. We provide 10 synthetic theorems in Section 3 of our supplementary material, each of which has been carefully evaluated by human experts. Some of them are deemed to meet IMO-level difficulty. Notably, one of these theorems, which is considered of medium difficulty, was selected as a problem for a major city’s 2024 Mathematical Olympiad.
>
> We will also generate a large test set with hierarchical difficulty levels, some of which will reach IMO-level. This dataset will be made available on the official AIPS website: https://sites.google.com/view/aips2.
>
> ## The Architectures of the Value Network in the Neural Prover
> The architecture of the value network is detailed in Section 2.2 of the supplementary material. It consists of two primary components: a pre-trained transformer encoder, Llemma-7b [1], and a multilayer perceptron (MLP) with layers of size 4096×256×1. The MLP outputs a value within the interval (0,1).
>
> ## The Difference Between “AIPS with Pretrained Value Network” and “AIPS”. What Data is the Network Pre-trained On and How Does It Differ from 'AIPS'?
> Both “AIPS with pretrained value network” and “AIPS” have a value network. The value network of “AIPS” is fine-tuned based on the pretrained value network from “AIPS with pretrained value network”.
> ### Pretraining Stage:
> We pretrain the value network on 191,643 theorems generated by AIPS’ deductive engine, using the tree-depth heuristic function as the ground truth. The tree-depth of an inequality is the maximum depth of the expression trees on both sides. We discuss the pretraining process in Section 3.3.2 of our paper.
> ### Fine-Tuning Stage:
> AIPS further improves its performance by solving problems in a filtered dataset and fine-tuning the value network. This filtered dataset, a subset of the generated dataset, contains 1,000 problems sorted from easy to hard. After fine-tuning, AIPS solves 10 out of 20 problems in the test set and demonstrates increased efficiency by finding proofs with fewer search loops. (Please refer to Sections 2.2 and 2.3 of our supplementary material for more details.)
> ## Pattern Matching of Theorems with Expression Trees in Figure 1, and the Representations of the Theorems During Pattern Matching, e.g., AM-GM Inequality
> Pattern matching within an expression tree involves identifying structures that match one side of an inequality theorem. These matched structures are then substituted into the other side of the theorem using the corresponding parameters from the tree. While performing pattern matching, the theorem is represented as a tree-structure.
>
> ### Explanation of Pattern Matching in Figure 1
> Theorem (AM-GM): for non-negative real numbers $a_1,a_2,…,a_n​, a_1+a_2+⋯+a_n≥n(a_1 a_2…a_n)^{1/n}.$
>
> In Figure 1(a), as the deductive engine traverses the expression tree of $xyz/(x+y+z)$​, it identifies that the node $x+y+z$ is an add-type expression, matching the left-hand side of AM-GM. The engine then generates new inequalities using AM-GM, such as $x+y+z≥3(xyz)^{1/3}$ and $x+y+z≥2((x(y+z))^{1/2}$​. It also deduces that the entire expression increases as the value of $x+y+z$ decreases, resulting in new inequalities like $xyz/(x+y+z)≤(xyz)^{2/3}/3$​ and $xyz/(x+y+z)≤xyz/(2(x(y+z))^{1/2})$​. We apologize for the typos in the obtained inequalities at the bottom of Figure 1(a) and will correct them in the final version.
>
> In Figure 1(b), we show that pattern matching at different nodes can generate inequalities in different directions. Matching the root with AM-GM results in inequalities like $1/(a+b)+1/(b+c)+1/(c+a) \geq 3/((a+b)(b+c)(c+a))^{1/3}$, while matching AM-GM at other nodes yields upper bounds such as $1/(2\sqrt{ab})+1/(2\sqrt{bc})+1/(2\sqrt{ca})$.
>
> We discuss pattern matching in Section 3.1.2 of our paper and in Section 1.2.2 of the supplementary material.
> ## Explanation of Sets 'S' and 'O' in Algorithm 1 and where they are obtained
> The set 'S' refers to a collection of inequality theorems, including the AM-GM inequality, the weighted AM-GM inequality, Hölder’s inequality, Jensen’s inequality, Schur’s inequality, and Muirhead’s theorem.
>
> The set 'O' comprises the transformation rules for manipulating inequalities. Here are some examples:
>
> - **zero_side**: subtract one side from the other to make one side of an inequality equal to zero
> - **no_sep_denom**: combine fractions on both sides of an inequality
>
> (Please refer to Section 1.2.1 of the supplementary material for more examples.)
>
> Both 'S' and 'O' are written by the authors and integrated into the AIPS deductive engine.
> ## Limitations
> AIPS currently relies on handwritten theorems and matching rules, which can be time-consuming to create. More efforts will be devoted to automatically discovering these rules. Additionally, the number of theorems in AIPS' deductive engine is limited. Expanding this collection, along with operational rules, would enhance AIPS' ability to solve more complex problems and discover more valuable theorems.
>
> [1] https://huggingface.co/EleutherAI/llemma_7b

---

> > ### Comment · Reviewer_gDey · 2024-09-01
> >
> > I apologize for the late response, but thanks for the answer! I am satisfied by the answer. I would additionally appreciate more details on the pattern matching algorithm in the actual body of the paper (the current writeup submitted is a bit vague but the rebuttal clarified it), space permitting. Also for the other clarifications, if you could add a sentence or two for each in the main paper, that would be great.

---

> > > ### Author Response · Authors · 2024-09-01
> > >
> > > Thank you for your constructive feedback! We’re pleased to hear that our clarifications addressed your concerns.
> > > We will definitely add more details about the pattern matching process and the other points you mentioned to further improve the overall clarity in the final version.

---

### Author Response · Authors · 2024-09-01

Dear Reviewers,

We hope this message finds you well. We sincerely appreciate the time and effort you’ve dedicated to reviewing our paper. Throughout the discussion, we have carefully addressed all the concerns and questions raised. However, as the discussion process comes to a close, we have not yet received any further responses or engagement, except from Reviewer V585.

With limited time remaining, we kindly request that you review our rebuttal and provide any final feedback.

Thank you for your time and attention.

Best regards,

Authors

---

### Decision · Program_Chairs · 2024-09-26

**Decision:**

Accept (Poster)

**Comment:**

This paper introduces a novel Algebraic Inequality Proving System (AIPS), which combines symbolic and neural reasoning to tackle Olympiad-level algebraic inequalities, a domain that has been largely unexplored in AI-aided theorem proving. The integration of a synthetic theorem generator, symbolic deductive engine, and neural proof search makes this work stand out. AIPS addresses key challenges in generating high-quality theorem datasets and establishes a benchmark by solving 10 out of 20 International Mathematical Olympiad (IMO) problems. This marks a significant advancement in AI-driven automated reasoning, especially in algebraic inequalities where the search space is large and requires deep reasoning.

The paper is technically sound and brings novel contributions, including the use of a value network trained with curriculum learning to guide the proof search, significantly improving performance. The authors also successfully rebutted concerns regarding dataset size, noting the difficulty in sourcing Olympiad-level problems but promising expansions through synthetic generation and additional sources. Despite minor concerns about test set size and the need for more clarity on value network fine-tuning, the system's ability to solve challenging inequality problems autonomously without human demonstrations makes this work of high significance. I recommend acceptance due to its originality, clear methodological contributions, and potential impact on the AI4Math community.